# Surface restructuring of a perovskite-type air electrode for reversible protonic ceramic electrochemical cells

Kai Pei[1,7], Yucun Zhou[2,7], Kang Xu[1,7], Hua Zhang[1], Yong Ding[2], Bote Zhao [1], Wei Yuan[2,3], Kotaro Sasaki[4], YongMan Choi [5✉], Yu Chen [1,6✉] & Meilin Liu [2✉]

Reversible protonic ceramic electrochemical cells (R-PCECs) are ideally suited for efficient energy storage and conversion; however, one of the limiting factors to high performance is the poor stability and insufficient electrocatalytic activity for oxygen reduction and evolution of the air electrode exposed to the high concentration of steam. Here we report our findings in enhancing the electrochemical activity and durability of a perovskite-type air electrode, $Ba_{0.9}Co_{0.7}Fe_{0.2}Nb_{0.1}O_{3-\delta}$ (BCFN), via a water-promoted surface restructuring process. Under properly-controlled operating conditions, the BCFN electrode is naturally restructured to an Nb-rich BCFN electrode covered with Nb-deficient BCFN nanoparticles. When used as the air electrode for a fuel-electrode-supported R-PCEC, good performances are demonstrated at 650 °C, achieving a peak power density of 1.70 W cm$^{-2}$ in the fuel cell mode and a current density of 2.8 A cm$^{-2}$ at 1.3 V in the electrolysis mode while maintaining reasonable Faradaic efficiencies and promising durability.

[1] School of Environment and Energy, South China University of Technology, Guangzhou 510006, China. [2] School of Materials Science and Engineering, Georgia Institute of Technology, Atlanta, GA 30309, USA. [3] School of Mechanical and Automotive Engineering, South China University of Technology, Guangzhou 510640, China. [4] Chemistry Department, Brookhaven National Laboratory, Upton, NY 11973, USA. [5] College of Photonics, National Yang Ming Chiao Tung University, Tainan 71150, Taiwan. [6] Guangdong Provincial Key Laboratory of Atmospheric Environment and Pollution Control, South China University of Technology, Guangzhou Higher Education Mega Centre, Guangzhou 510006, China. [7] These authors contributed equally: Kai Pei, Yucun Zhou, Kang Xu. ✉email: ymchoi@nctu.edu.tw; eschenyu@scut.edu.cn; Meilin.liu@mse.gatech.edu

Reversible protonic ceramic electrochemical cells (R-PCECs) are highly efficient energy conversion and storage devices that can be operated at intermediate temperatures (e.g., 300–700 °C), thus holding great promise as a game-changer for the widespread utilization of intermittent renewable energy[1–3]. The kinetics of the electrochemical reactions on the air electrode, oxygen reduction reaction (ORR) in the fuel cell (FC) mode and oxygen evolution reaction (OER) in the electrolysis (EL) mode, are complex and notoriously sluggish at low temperatures. The polarization resistance of the air electrode still dominates the energy loss of R-PCECs. Thus, breakthroughs in the development of highly efficient and durable air electrode materials and structures are urgently needed to advance the R-PCEC technology.

The porous air electrode has to allow rapid transport of steam and $O_2$ to or away from the active sites for electrode reactions and facilitate fast charge transfer associated with ORR and OER. The most important properties of a high-performance air electrode material are high conductivities of electrons, oxygen ions, and protons, high electrocatalytic activity for ORR and OER, and sufficient compatibility with other cell components. While many good air electrode materials have been developed for solid oxide cells based on oxygen ion conductors, including $La_{0.6}Sr_{0.4}Co_{0.2}Fe_{0.8}O_{3-\delta}$ (LSCF)[4], $Ba_{0.5}Sr_{0.5}Co_{0.8}Fe_{0.2}O_{3-\delta}$ (BSCF)[4], and $La_{0.8}Sr_{0.2}CoO_{3-\delta}$ (LSC)[5], they are proven unsuitable for R-PCECs due to limited proton conductivity and poor stability against high concentration of steam. Recently, significant efforts have been devoted to the development of triple conducting oxides (TCO): mixed conductors that allow fast transport of protons, oxygen ions, and electrons. Among the TCO materials studied, double perovskite oxides exhibit much faster chemical diffusion and surface exchange than the conventional $ABO_3$-type perovskite oxides, leading to lower polarization loss and better cell performance[6–11]. However, these materials often contain a considerable amount of expensive elements (such as praseodymium or cobalt) and potentially have a high thermal expansion coefficient. The use of these double perovskite electrodes may increase the capital cost and the risk of delamination during thermal cycling. In addition, exposure to high partial pressures of steam may result in rapid degradation in the catalytic activity of these materials, due likely to the formation of Sr/Ba/Co-enriched surface or particles[12–14]. Therefore, a good air electrode for R-PCECs must also have sufficient tolerance against high concentrations of steam in the electrolysis mode[15]. To date, considerable efforts have been devoted to the development of highly active and durable air electrode materials/structures for R-PCECs[3,6,11,16–19]. One effective method to enhance the surface stability of electrodes is the surface modification with a series of effective catalyst coatings via a low-cost infiltration process[20–23]. The coatings have dramatically enhanced the ORR activity and durability of the electrode of oxygen-ion-based SOFCs by increasing the surface oxygen exchange kinetics and inhibiting the surface Sr-segregation. However, surface modification may be ineffective for enhancing the durability of R-PCECs air electrodes since the conditions are much harsher in the electrolysis mode, where the steam content can be >50%. Thus, it is crucial to enhance the intrinsic stability of the air electrode when exposed to a high concentration of steam. For instance, an air electrode consisting of Ca-doped $PrBa_{0.8}Ca_{0.2}Co_2O_{5+\delta}$ (PBCC) has demonstrated a good electrochemical activity and acceptable stability in a high-steam concentrated air[3,24]. It is suggested that PBCC possibly reacted with water to generate nano-sized $BaCoO_{3-\delta}$ (BCO) on the surface, forming a BCO coated PBCC electrode[3]. The enhanced electrocatalytic activity and stability were due most likely to the synergistic effect of the unique attributes of BCO (for rapid water dissociation) and PBCC (for fast associative oxygen desorption)[3]. However, the stability of Co-based perovskite oxides and more expensive

Pr material (compared with Ba) could be a potential concern under realistic operating conditions for further commercialization. Some of the well-developed air electrode materials are derived from BCO perovskite oxides, such as BSCF[25], $PrBa_{0.5}Sr_{0.5}Co_{2-x}Fe_xO_{5-\delta}$ (PBSCF)[26], and $NdBa_{1-x}Ca_xCo_2O_{5+\delta}$ (NBCC)[27], which are reported to show high ORR activity at reduced temperatures. High-valence cations such as Nb have been considered as the most effective dopant at the B-site for improving the phase/lattice stability and oxygen permeability of the Co-based perovskite oxides[28]. For instance, Zhou et. al. reported a novel $SrSc_{0.175}Nb_{0.025}Co_{0.8}O_{3-\delta}$ cathode material with a rapid bulk oxygen diffusion rate and a good cell performance at 500 °C[26]. Further, they reported another material with high-valence dopants, $SrCo_{0.8}Nb_{0.1}Ta_{0.1}O_{3-\delta}$, which showed an even higher performance at 500 °C[29].

Inspired by these developments, here we report a perovskite material with A-site deficiency, $Ba_{0.9}Co_{0.7}Fe_{0.2}Nb_{0.1}O_3$ (BCFN), which demonstrates high ORR/OER activity and promising durability under typical R-PCECs operation conditions. It is observed that BCFN interacts with water, as confirmed by transmission electron microscopy (TEM) analyses, but fine Nb-deficient BCFN nanoparticles (NPs) on the BCFN surface are formed, providing more reaction sites and outstanding durability for oxygen reduction/evolution or water electrolysis. When the BCFN air electrode is tested in a $BaZr_{0.1}Ce_{0.7}Y_{0.1}Yb_{0.1}O_{3-\delta}$ (BZCYYb)-based single cell with a configuration of Ni-BZCYYb | BZCYYb | BCFN, the cell delivers high performances in both fuel cell mode (e.g., a typical peak power density of 1.7 W cm$^{-2}$ at 650 °C) and electrolysis mode (e.g., a current density of 2.8 A cm$^{-2}$ at 1.3 V for water electrolysis).

## Results

As schematically shown in Fig. 1, in the water electrolysis (EL) mode, steam is fed to the air electrode and pure hydrogen is only generated at the fuel electrode. No $H_2$ purification is needed, thus decreasing the complexity and cost of the system. The nanoparticles (NPs) on the BCFN surface, exsoluted from the parental electrode, are confirmed to be Nb-deficient BCFN nanoparticles by TEM and X-ray diffraction (XRD) analyses. Furthermore, since the Ni-based fuel electrode is only exposed to dry hydrogen, the active Ni-phase no longer has the risk of Ni oxidation in the

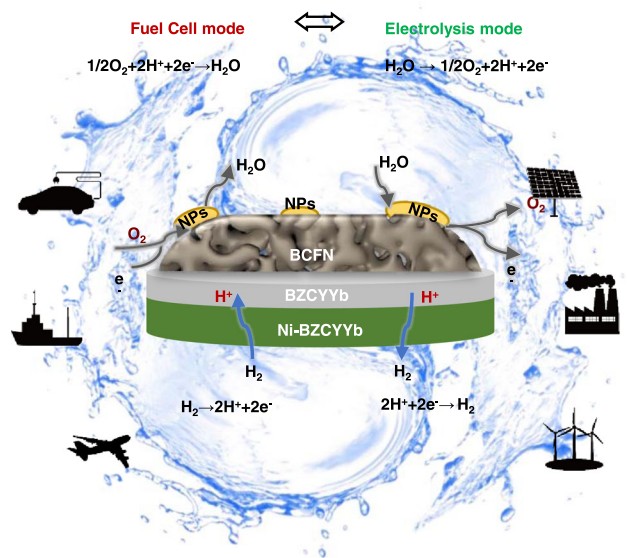

**Fig. 1 Reactions on an R-PCEC and their applications.** Schematic illustration of an R-PCEC with a steam-induced surface restructured BCFN air electrode operated in both FC and EL modes.

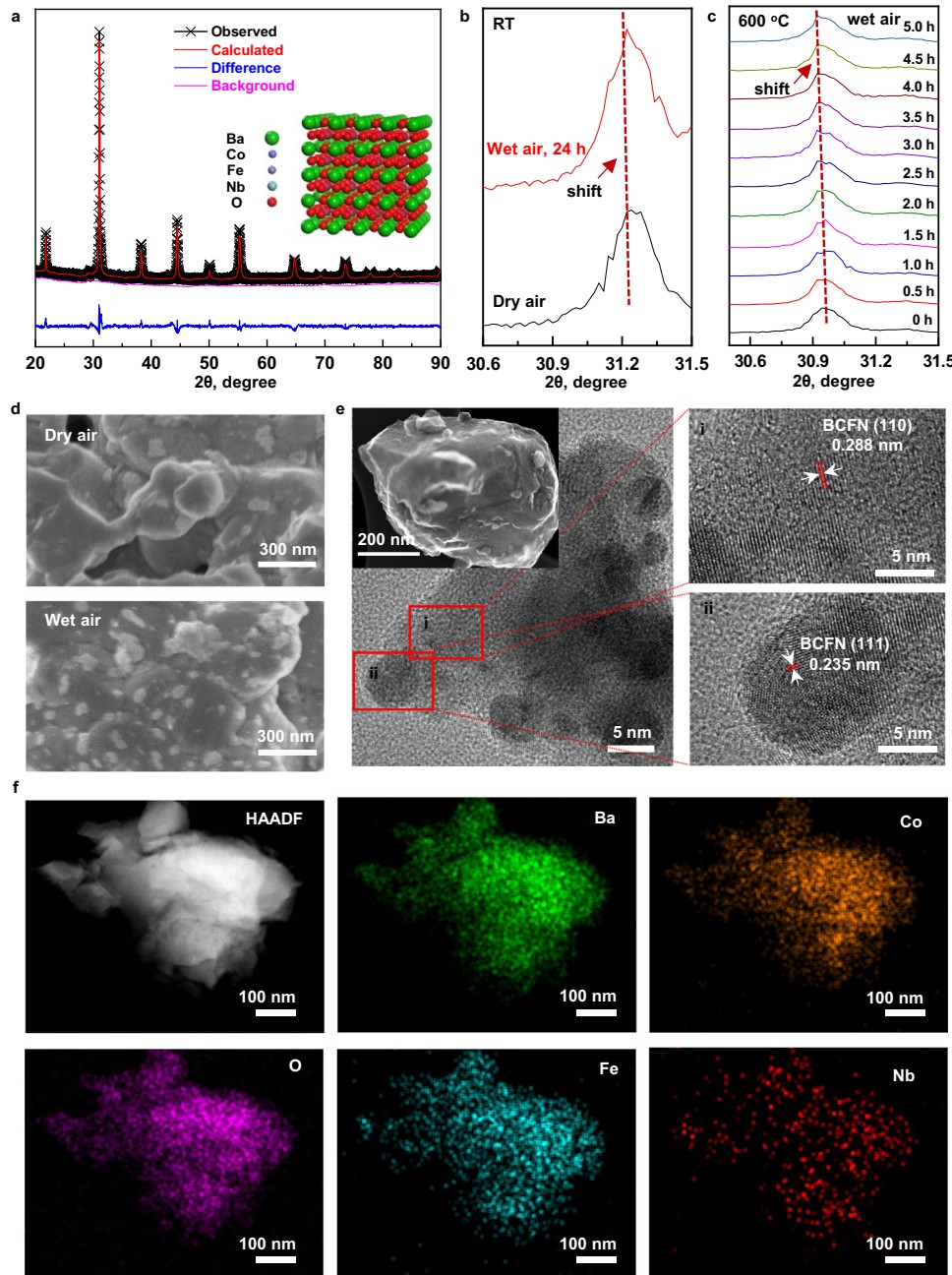

**Fig. 2 Structural characterization of BCFN powder. a** A Rietveld refinement of XRD patterns for a BCFN powder with measured data (black dots), simulated (red line; calculated profile), background (pink line), difference curves (blue line), and a schematic of a BCFN crystal structure (inset). The powders were calcined at 1100 °C for 2 h in the air; **b** Room temperature XRD patterns of the BCFN powders after a treatment at 650 °C in dry air and wet air (24 h, 3% $H_2O$); **c** XRD patterns (collected at 600 °C) of the BCFN powders treated in wet air; **d** SEM images of BCFN powders before (top) and after (bottom) exposure to the wet air (3% $H_2O$) at 650 °C for 2 h; **e** High-resolution TEM images of BCFN grains, including BCFN substrate and BCFN NPs; **f** High-angle annular dark-field STEM image from the BCFN powders, and the X-ray energy dispersive spectrum mapping of Ba, Co, O, Fe, and Nb.

electrolysis mode and the R-PCECs can achieve high fuel utilization and efficiency in the fuel cell (FC) mode;[30] in contrast, fuel dilution is unavoidable in an oxide-ion conductor based cell[31].

Shown in Fig. 2a are the XRD patterns of BCFN with no detectable impurities. An XRD refinement of the BCFN crystal structure displayed a typical perovskite structure ($Pm\bar{3}m$), with a reasonable reliability fitting factors of 1.83. It is noted that no impurities were detected in the mixture of BZCYYb and BCFN (fired at 1000 °C for 2 h in the air), indicating good chemical compatibility between BZCYYb and BCFN (Supplementary Fig. 1). Slight shifts of the main peak of the BCFN sample were

observed (Fig. 2b and Supplementary Fig. 1), an indicator of a slight structure change (likely a lattice expansion) after being treated at 650 °C in wet air. The structure expansion induced by steam is further confirmed by the high-temperature in situ XRD examination of BCFN powder in wet air at 600 °C (Fig. 2c and Supplementary Fig. 2). After interacting with water, parental BCFN grains developed numerous nanoparticles on the surface, as suggested by the scanning electron microscopy (SEM) images of BCFN samples treated in dry and wet air (Fig. 2d). Shown in Fig. 2e are the high-resolution TEM images of BCFN powders, treated in the air with 3% $H_2O$ for 24 h. The lattices with spacing

distances of 0.288 nm and 0.235 nm are corresponding to the (110) and (111) facets of BCFN, respectively, which may suggest that the NPs on the surface of BCFN grain are still BCFN, but with a slight difference in composition, as elaborated later. This is further confirmed by the high-angle annular dark-field scanning transmission microscopy (HAADF-STEM) image of BCFN powder (treated in the air with 3% $H_2O$ for 24 h), and the X-ray energy dispersive spectrum mapping of Ba, Co, O, Fe, and Nb. All these analyses strongly suggest that the surfaces of BCFN powder are restructured into BCFN grains covered with fine NPs under a typical operation condition for R-PCECs.

Such a self-restructured electrode demonstrates an encouraging good electrochemical activity and durability when applied as an air electrode of R-PCECs in dual-mode of FC and EL. The BCFN electrode showed a low area-specific resistance (ASR) when compared with other cathodes[32–44] tested at intermediate temperatures (from 700 to 500 °C) in humidified air, as displayed in Fig. 3a and Supplementary Table 1. It is noted in Supplementary Fig. 3 that BCFN air electrodes show the lowest polarization resistance ($R_p$) of ~0.09 Ω cm² (at 650 °C), determined from

symmetrical cells, compared with composite electrodes (BCFN: BZCYYb = 7:3 and 5:5); it may suggest that BCFN alone showed a sufficient reaction activity. Therefore, in the following single-cell test, BCFN alone was used as the electrode material. Shown in Fig. 3b is a typical cross-sectional SEM image of a fuel-electrode supported single-cell with the BCFN air electrode after electrochemical testing in FC and EL modes. As shown in Supplementary Fig. 4, the multiple layers of the cell are well bonded together (no cracks or delamination), including a porous Ni-BZCYYb anode supporting layer (ASL, ~600 μm), a porous Ni-BZCYYb anode functional layer (AFL, ~25 μm), a dense BZCYYb electrolyte layer (~10 μm), and a porous BCFN layer (~15 μm). The AFL has finer pores and a larger surface area (due mainly to the reduction of NiO to metallic Ni) than the ASL, providing more triple-phase boundaries for electrochemical reactions. In contrast, the ASL has larger pores and continuous channels (due mainly to the removal of the pore-former), providing facile paths for gas transport. The inset of Fig. 3b shows the rough morphology of the BCFN surface, consistent with the observations in Fig. 2d. Shown in Fig. 3c are the typical *I-V-P* curves of the cell

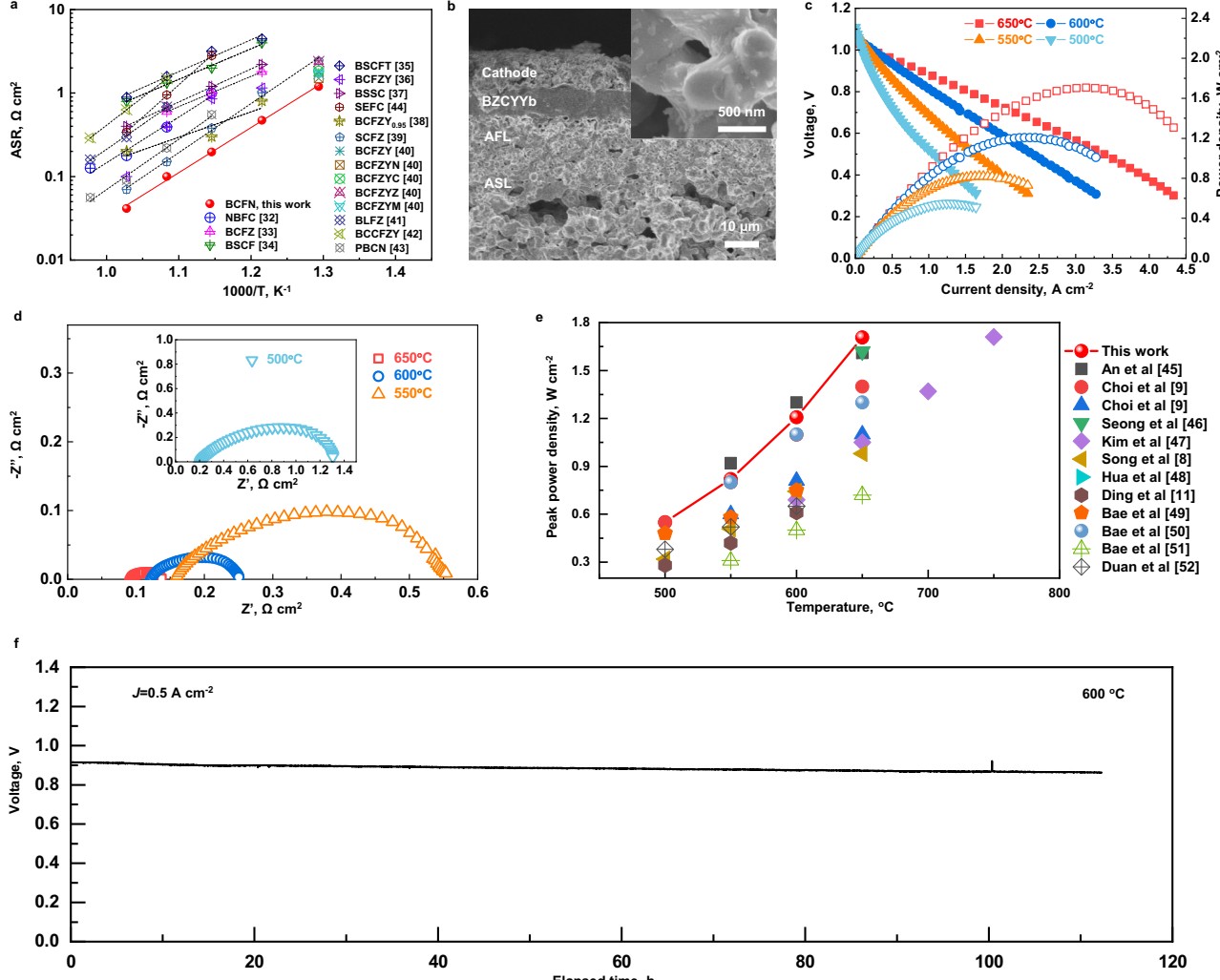

**Fig. 3 Electrochemical performance of R-PCECs with a BCFN air electrode in the FC mode. a** Temperature dependence of the polarization resistance ($R_p$) of a symmetrical cell with BCFN and other high-performance electrodes[32–44], as shown in Supplementary Table 1; **b** A cross-section image of a Ni-BZCYYb fuel electrode supported single cell. Inset is the detailed SEM image of cathode; **c** Typical *I-V-P* curves of the single-cell measured at 650, 600, 550, and 500 °C, respectively; **d** Typical *EIS* curves of the single cell under OCVs using $H_2$ (3% $H_2O$) as the fuel and ambient air as the oxidant, measured at 650, 600, 550, and 500 °C, respectively; **e** The comparison of the peak power densities of PCFCs with different cathode materials[8,9,45–53], and BCFN (this work), tested from 800 to 500 °C; and **f** A short-term stability test of the single cell at a constant current density of 0.5 A cm$^{-2}$ and 600 °C.

measured in the FC mode at 650–500 °C using wet $H_2$ (3% $H_2O$) as the fuel and ambient air as the oxidant. The cell with the BCFN air electrode shows high peak power densities ($P_{max}$) of 1.7, 1.2, 0.8, and 0.5 W cm$^{-2}$ at 650, 600, 550, and 500 °C, respectively. Figure 3d displays the typical electrochemical impedance spectra (*EIS*) spectra of the cell measured at 650–500 °C under open-circuit voltage conditions. The cell has a polarization resistance of only 0.04 Ω cm$^2$ at 650 °C. As summarized in Fig. 3e, the $P_{max}$ of our cells surpasses most of the performances reported for protonic ceramic fuel cells under similar operating conditions (Supplementary Table 2)[7–9,11,45–52]. Moreover, the cell with the BCFN air electrode shows good stability in the FC mode, maintaining a voltage of 0.9 V at 0.5 A cm$^{-2}$ and 600 °C for an operation period of over 100 h (Fig. 3f).

In addition to the fuel cell tests, R-PCECs using the BCFN air electrodes were also evaluated under an electrolysis cell mode. Figure 4a shows the current-voltage (*I–V*) curves of the cell collected from 1.4 V to OCV when the fuel electrode was exposed to humidified hydrogen (3% $H_2O$) and the air electrode was supplied with humidified air (3% $H_2O$). Typical high current

densities of ~2.8, 1.6, 0.7, and 0.3 A cm$^{-2}$ at a cell voltage of 1.3 V were achieved at 650, 600, 550, and 500 °C, respectively. Shown in Fig. 4b is the comparison of current densities from electrolysis cells at a cell voltage of 1.3 V at different operation temperatures (Supplementary Table 3)[2,6,7,53–56]. It is suggested that our cell performance based on BCFN-based air electrodes is among the top, especially at 650 °C. Shown in Fig. 4c is the long-term stability of the cell with the BCFN air electrode in the EL mode at a constant current density of 1 A cm$^{-2}$ (for over 200 h) and 0.5 A cm$^{-2}$ (for ~380 h), respectively, at 550 °C. Furthermore, we operated a cycling test; the cell was running in dual modes of FC and EL. Figure 4d shows the cell voltage as a function of the time when the R-PCEC was operated at 650 °C between the modes of EC and EL (2, 4, 8, 12, and 1 h for each mode) at ± 0.5 A cm$^{-2}$. The cell displayed a degradation of less than 1% within 240 h and 51 cycles (Fig. 4d), demonstrating the good durability and reversibility of the cells with the BCFN air electrodes. Faradaic efficiency describes the efficiency with which electrons participate in the desired reaction in an electrochemical system. For the electrochemical hydrogen evolution reaction, Faradaic efficiency

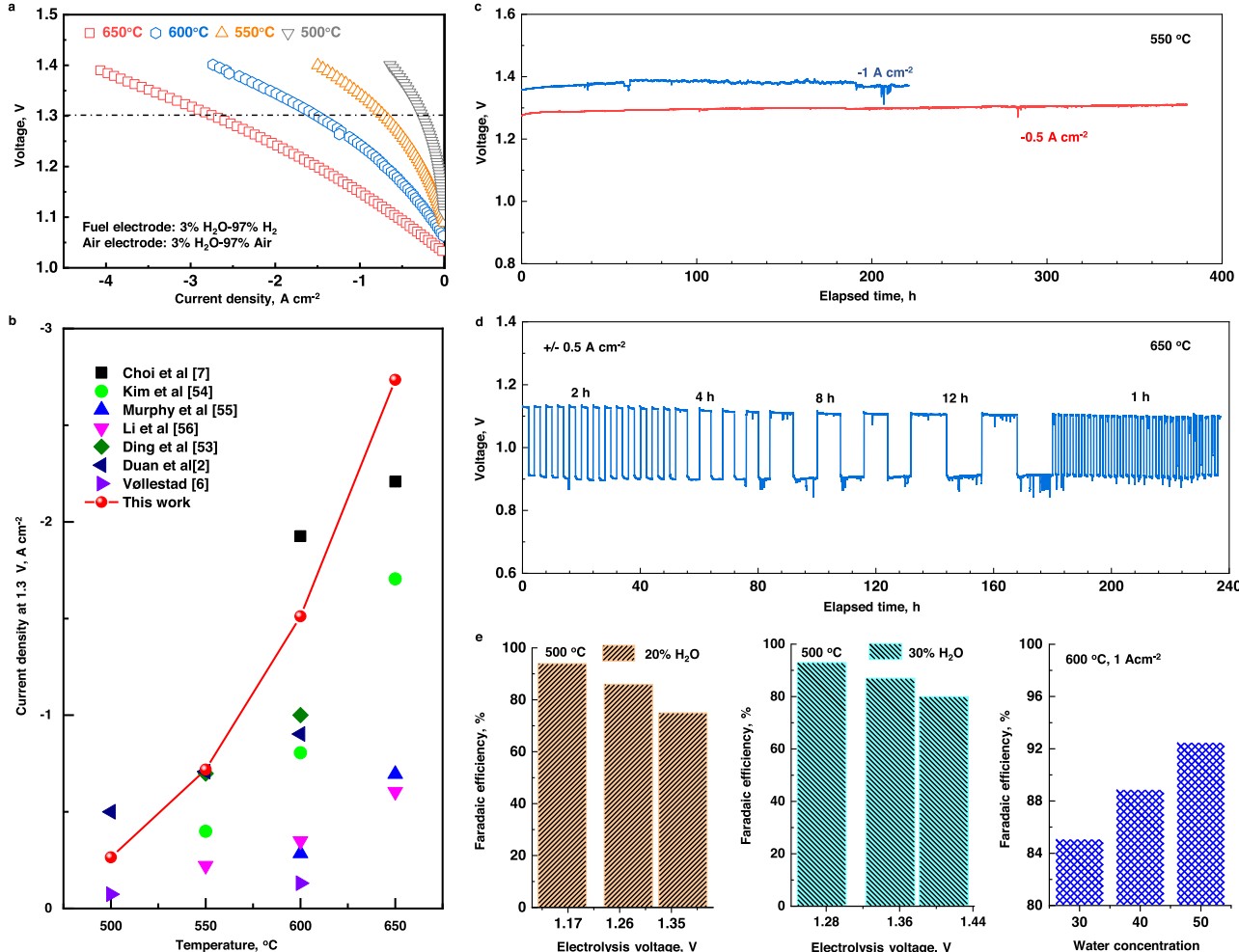

**Fig. 4 Electrochemical performance of R-PCECs with the BCFN air electrode in the EL mode. a** Typical *I-V* curves of the R-PCECs measured at 650, 600, 550, and 500 °C, respectively with humidified $H_2$ (3% $H_2O$) in the fuel electrode and humidified air (3% $H_2O$) in the air electrode in the EL mode; **b** the comparison of current densities of different air electrode materials reported and this work at 1.3 V from 650 °C to 500 °C;[2,6,7,53–56] **c** Stability test of the single cell at current densities of −0.5 A cm$^{-2}$ and −1.0 A cm$^{-2}$ at 550 °C; **d** Reversible operation of the R-PCEC: the cell voltage as a function of time when the operating mode was switched between the modes of FC and EL at a current density of ±0.5 A cm$^{-2}$ for intervals of 2, 4, 8, 12, and/or 1 h for each mode at 650 °C; and **e** Faradaic efficiencies of R-PCECs for producing hydrogen at different electrolysis voltages, and different $H_2O$ concentration (20% and 30%) in air at 500 °C, and Faradaic efficiencies of the cells when the air electrode was operated in different water concentration at a current density of 1 A cm$^{-2}$ at 600 °C.

is defined as the ratio of the actual amount of $H_2$ produced by the electrolysis cell to the predicted amount of $H_2$ from the current passing through the cell using Faraday's law of electrochemical equivalence. Shown in Fig. 4e are the variations in Faradaic efficiency for producing hydrogen at different current densities when the air electrode was exposed to humidified air with 20% and 30% of steam at 500 °C. With the increase in cell voltages, the Faradaic efficiency was reduced. In addition, when the current density was 0.5 A cm$^{-2}$ (with a cell voltage of 1.3 V), the Faradaic efficiency was higher for the humidified air with a higher concentration of steam. It is also consistent with the performance of cells operated at 1 A cm$^{-2}$ current density and 600 °C. When the water concentration was changed from 30% to 40% and 50%, the Faradaic efficiency was increased from ~85% to ~92% (as shown in the right figure of Fig. 4e).

## Discussion

To gain deeper insights into the demonstrated activity and durability of the BCFN air electrode, the surface morphology of air electrodes before and after the durability tests were characterized using TEM and energy-dispersive X-ray spectroscopy (EDS). Shown in Supplementary Fig. 5 are a STEM image and elemental mapping of Ba, Co, Fe, and Nb cations of the BCFN samples before the electrochemical test, suggesting that each element is uniformly distributed on the surface of the air electrode. After the testing, however, exsoluted BCFN nanoparticles were observed on the surface of BCFN, as shown in Fig. 5a. An elemental mapping of Nb (Fig. 5a) and a detailed elemental profile along the scanning line shown in Fig. 5b suggest that the surface nanoparticles are likely Nb-deficient BCFN while the substrates are Nb-rich BCFN (Supplementary Figs. 5–7). BCFN electrodes with different Nb content were further evaluated in symmetrical cells. The results indicated that BCFN with less Nb shows a relatively higher activity, acquired from both symmetrical cells and full cells (Supplementary Figs. 8–11). The in situ formed Nb-deficient BCFN NPs on BCFN electrodes are expected to exhibit better structural stability since the risk of agglomeration is greatly reduced, thus enhancing the durability and thermal stability of the electrode[57–60].

To elucidate the formation mechanism of Nb-deficient BCFN NPs on the BCFN air electrode, bulk and surface properties of various Ba-containing perovskites were studied. Stoichiometric $BaBO_3$ (B = Co, Fe, or Nb)[61] with a cubic structure ($Pm\bar{3}m$) was first optimized (Supplementary Table 4 and Supplementary Fig. 12). Then the 16-layered BO-terminated (001) surface was applied to examine the surface energy that can be used to measure the surface stability[62] (Supplementary Fig. 13). According to the order of NbO (0.93 J m$^{-2}$) > FeO (0.82 J m$^{-2}$) > CoO (0.51 J m$^{-2}$), we assume the segregation of Nb atoms from the bulk to the surface would take place less favorably than Co and Fe. We also observed that the surface energy of their BaO-terminated surface depends on B cations (Supplementary Table 4). It follows the similar tendency of increasing in lattice parameters ($a_{0,\ DFT}$) in the order of Nb- (4.1208 Å) > Fe- (3.9761 Å) > Co-containing (3.9562 Å) unit cells. With this initial comparison, as summarized in Supplementary Table 5, we further carried out the segregation energy calculations for co-doped $Ba(B_{0.5}B'_{0.5})O_3$ (B, B' = Co, Fe, or Nb; Supplementary Fig. 13) with 50% dopant concentration. It demonstrates that the co-doping may be more practical for examining the NP formation phenomena than $BaBO_3$ as the bulk property is altered by mixing two dopant cations. This study considered the segregation energy for each B dopant cation[63], not a layer[64]. As displayed in Supplementary Fig. 13, 16-layered slab surface models of $Ba(B_{0.5}B'_{0.5})O_3$(001) (B, B' = Co, Fe, or Nb) were prepared. Then the segregation energy ($E_{segr}$) was calculated

on BO- and B'O-terminated (001) surfaces. We selected the ninth layer in the middle of the slab as the bulk position for $E_{B,bulk}$. Although numerous possibilities are available for static DFT-based simulations, we only considered the shortest segregation path from the ninth layer to the top-most layer (Supplementary Fig. 13). As shown in Supplementary Table 5 and Fig. 5c, the segregation energy follows the order of Nb > Fe > Co, demonstrating that Co and Fe more easily segregate from the bulk to the surface than Nb, resulting in Nb-deficient NPs. A-site deficient $Ba_{0.9}(Co_{0.63}Fe_{0.25}Nb_{0.13})O_{3.0}$ was then constructed to investigate the Nb-deficient NP formation in more detail. As summarized in Supplementary Table 6, the theoretical lattice constant of A-deficient $Ba_{0.9}(Co_{0.63}Fe_{0.25}Nb_{0.13})O_{3.0}$ ($a_{0,DFT} = 3.9768$ Å, BCFN) with a cubic structure ($Pm\bar{3}m$) (Supplementary Fig. 12) is in line with that from an experiment ($a_{0,expt} = 4.0585$ Å)[65]. The electronic structure of BCFN is summarized in Supplementary Fig. 14. The broadband of Co-O and Fe-O bonds is covalent, leading to a strong hybridization with O 2p[66]. The 3d orbitals of Co and Fe are close to the Fermi level, while that of Nb 4d is more positively shifted, augmenting better chemical stability of BCFN[66]. Based on the surface energy calculations (CoO-termination: 0.78 J m$^{-2}$ versus CoFeNbO-termination: 1.03 J m$^{-2}$, Supplementary Fig. 13), we prepared the eight-layered CoO-terminated BCFN(001) surface to examine the segregation energy of Co, Fe, and Nb atoms located in bulk. We also assumed that the segregation of Ba atoms may be less important than B cations as the surface energies of BaO-terminated surfaces rely on B cations (i.e., BaO-CoO-BaO-CoFeNbO- (0.95 J m$^{-2}$) and BaO-CoFeNbO-BaO-CoO- (0.91 J m$^{-2}$)). Similar to $Ba(B_{0.5}B'_{0.5})O_3$(001), we only considered the shortest segregation path from the sixth CoFeNbO-terminated layer to the top-most CoO-terminated layer. Accordingly, we assumed that the Co-O bond formation initiates the Nb-deficient BCFN nanoparticle growth rather than Fe-O or Nb-O bond formation. As it may result in Co-deficient surfaces, we then prepared the Co-deficient CoO-terminated BCFN(001) surface by removing one Co atom on the surface, followed by segregation energy calculations ($Co_{,bulk}$, $Fe_{,bulk}$, or $Nb_{bulk}$ to $V_{Co,surf}$) (Fig. 5e). As shown in Fig. 5e, Nb is not favorable to segregate to the surface compared to Co and Fe (1.18 eV versus 0.39 eV and 0.17 eV, respectively), leading to the segregation tendency of Co > Fe > Nb. As summarized in Supplementary Fig. 13d, the CoFeNbO layer has one more Fe atom with that of 2.33 eV. One can conjecture that the NP formation via the segregation process occurs locally, depending on the segregated atom's interaction with neighboring atoms and oxygen vacancies[63]. For examining the plausible NP configurations, as an extreme case, we constructed BaBO (1 Ba, 4 B, and 8 O atoms) NP models formed on BCFN(001) (Supplementary Fig. 15). It leads to the segregation energy of 2.54 eV (Co), 5.07 eV (Fe), and (10.50 eV), which are qualitatively in line with the segregation trend with bare surface models. We investigated if the deficient Nb configuration formed on the surface could affect other dopants' segregation energy with this detailed confirmation following the segregation order of Co > Fe > Nb. As shown in Supplementary Fig. 13, the Nb atom in bulk and the Co atom on the surface were swapped, effectively lowering the two Fe atom's segregation energies from 0.71 eV to 0.09 eV and from 2.33 eV to −0.04 eV. It designates that hardly segregated Nb atoms on the surface could boost the segregation of co-dopants (i.e., Fe cations) in bulk. We successfully elucidated the mechanism of the Nb-deficient BCFN NP formation on the A-site deficient BCFN air electrode by computing the segregation energy of co-dopant cations and interacting of electrodes with steam under simulated conditions (Fig. 5f and Supplementary Figs. 16–18).

In summary, we present a new approach for designing a perovskite-type air electrode for high-performance and durable reversible protonic ceramic electrochemical cells. Under

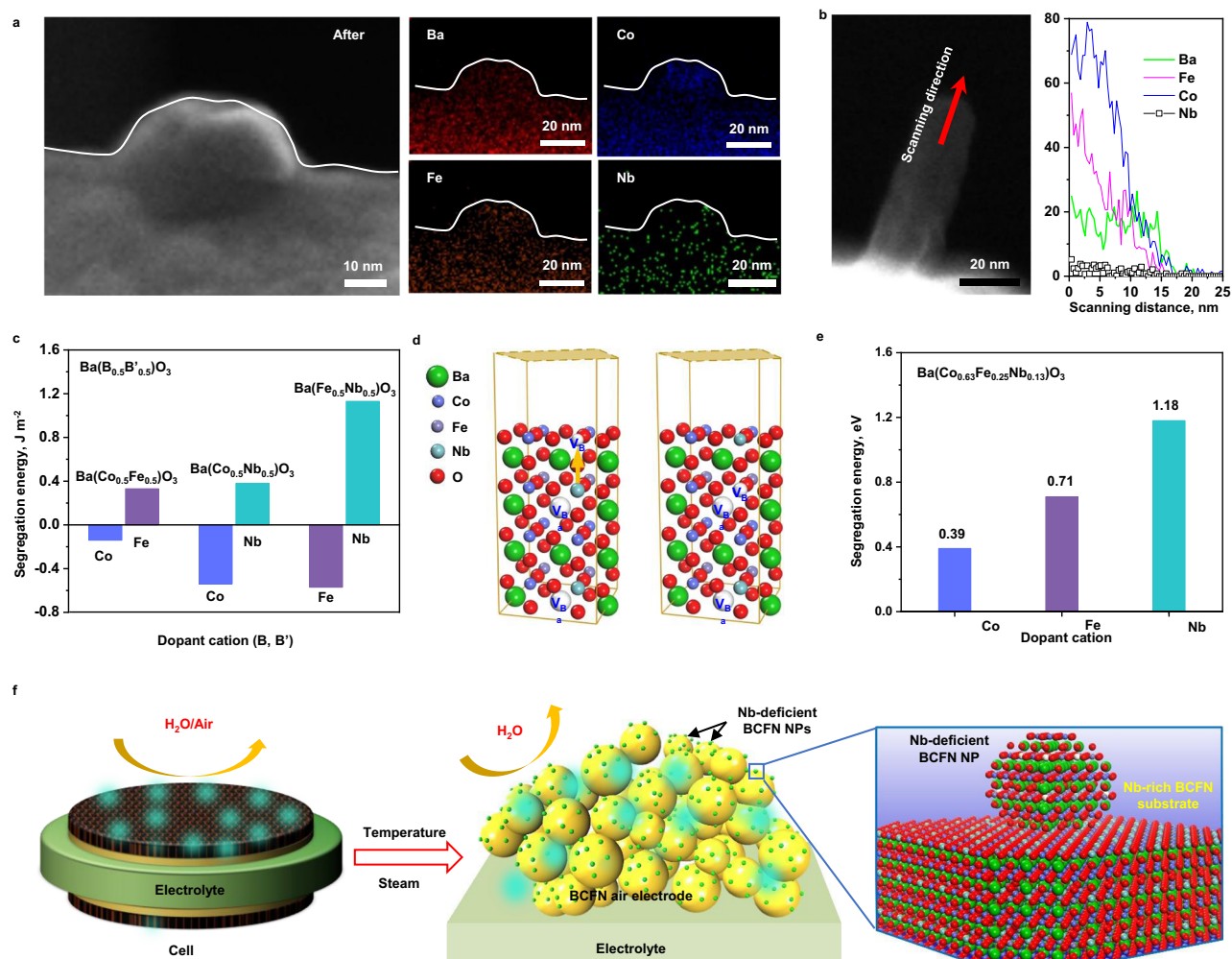

**Fig. 5 Formation of the high-performance air electrode of BCFN covered with fine BCFN NPs with less Nb. a**. TEM image and elemental (Ba, Co Fe, and Nb) mapping of a BCFN grain after electrochemical testing at 650 °C; **b**. Elemental profile along the scanning line shown in the TEM image; **c**. Comparison of computed segregation energies (J m$^{-2}$) for stoichiometric Ba(B$_{0.5}$B'$_{0.5}$)O$_3$ (B,B' = Co, Fe, or Nb); **d**. Illustration of CoO-terminated BCFN(001) surface models representing segregation of Nb from the bulk to the surface; **e**. Computed segregation energy (eV) for three different dopant cations of Ba$_{0.9}$(Co$_{0.63}$Fe$_{0.25}$Nb$_{0.13}$)O$_3$; and **f**. A schematic illustration of the formation of Nb-deficient BCFN NPs on the BCFN air electrode.

properly-controlled conditions, the BCFN electrode has been naturally restructured to an Nb-rich BCFN electrode covered with fine Nb-deficient BCFN NPs. The mechanism of the Nb-deficient BCFN NPs formation on the parental BCFN air electrode has been elucidated by computing the segregation energy of co-dopant cations using density functional theory (DFT) calculations. Cells with the BCFN air electrodes that we developed demonstrate high performances in both FC (e.g., a peak power density of 1.70 W cm$^{-2}$ at 650 °C) and EL (e.g., a current density of 2.8 A cm$^{-2}$ at 1.3 V for water electrolysis) modes. The demonstrated design of surface reconstructing is highly attractive not only for R-PCECs but also for other types of energy conversion and storage systems, including membrane reactors for the synthesis of high value-added chemicals.

## Methods

**Powder synthesis**. NiO powder used in this study is commercially available from H2-BANK. BaZr$_{0.1}$Ce$_{0.7}$Y$_{0.1}$Yb$_{0.1}$O$_{3-\delta}$ (BZCYYb) electrolyte materials were prepared via a reactive solid-state reaction. More details can be found in our early report[67]. Specifically, stoichiometrical BaCO$_3$, ZrO$_2$, CeO$_2$, Y$_2$O$_3$, and Yb$_2$O$_3$ (all purchased from Macklin) were mixed and then added to a high-energy ball-milling jar with a certain amount of ethanol for 24 h ball-milling. After being dried for 12 h, the precursor powders were then die-pressed into pellets at 10 MPa and calcined at 1100 °C for 12 h two times. The air electrode materials of A-site

deficient Ba$_{0.9}$Co$_{0.7}$Fe$_{0.2}$Nb$_{0.1}$O$_{3-\delta}$ (BCFN) and Ba$_{0.9}$Co$_{0.7}$Fe$_{0.2}$Nb$_x$O$_3$ ($x = 0.025$, and 0.05) were fabricated via a solution combustion method. More specifically, stoichiometrical Ba(NO$_3$)$_2$, Co(NO$_3$)$_2$·6H$_2$O, Fe(NO$_3$)$_3$·6H$_2$O, C$_4$H$_4$NNbO$_9$·nH$_2$O were dissolved into deionized water. Complexing agents of glycine and citric acid were then added sequentially into the solution with a molar ratio of total metal ions: glycine: citric acid at 1: 0.75: 0.75. After being heated and stirred at 90 °C for 2 h, the gel was moved to an oven and heated at 300 °C to complete the combustion process. Before making the electrode slurry, the ash was heat-treated in the air at 900 °C for 2 h[67].

**Fabrication details of symmetrical and fuel electrode supported cells**. Dense BZCYYb electrolyte pellets for symmetrical cells were fabricated by uniaxially pressing the BZCYYb powder (with 1% NiO as a sintering aid) and followed by sintering at 1450 °C for 5 h. Symmetrical cells with a configuration of BCFN| BZCYYb|BCFN were prepared via screen-printing the air electrode slurry onto both sides of the BZCYYb electrolyte. The BCFN screen-printing paste was prepared by mixing BCFN powder, ethyl cellulose, and terpinol with a weight ratio of 1: 0.04: 0.76 after ground for one hour in an agate mortar[67]. The symmetrical cells were then fired in the air at 950 °C for 2 h.

Fuel electrode-supported half-cells were lab-prepared via a three-layer co-tape-casting process. The slurries of BZCYYb electrolyte, NiO-BZCYYb fuel electrode functional layer, and NiO-BZCYYb fuel electrode supporting layer were sequentially cast onto a polymer film every 30 mins. After being sufficiently dried in ambient air at room temperature for ~15 h, the green tape (before firing) was punched into several pellets with a diameter of ~15 mm and fired at 600 °C for 2 h at a firing rate of 0.5 °C per minute. After that, the pre-fired pellets were then heated at 1,450 °C for 5 h (to densify the electrolyte layer) to form fuel electrode-supported half cells. The BCFN and Nb-deficient BCFN air electrodes were then

screen-printed onto the electrolyte of half cells. The full cells were then fired in the air at 950 °C for 2 h.

**Characterization and cell testing**. The XRD patterns of the powders of BCFN and BZCYYb were collected from a Bruker D8 advance (Germany Bruker) instrument under Cu Kα radiation ($\lambda = 0.15406$ nm). The morphology of powders and cells was detected by scanning electron microscopy (SEM, Hitachi SU8010, Japan). The fuel cells' performance was evaluated by testing cells with 3% $H_2O$ humidified hydrogen in the fuel electrode and ambient air in the air electrode. The electrolysis cell performance was evaluated by testing cells with 3% $H_2O$ humidified hydrogen in the fuel electrode and 3% $H_2O$ humidified air in the air electrode. The humidity of 3% is controlled by flowing the gas through a water-bubbler at room temperature. For most of the tests, button cells with an effective area of 0.28 cm$^2$ were used. For the Faradaic efficiency test, 1-inch cells with an effective area of 2 cm$^2$ were used to increase the accuracy of the measurement. For the fuel cell (small button cell) test, the fuel electrode was exposed to 20 mL min$^{-1}$ humidified $H_2$ (3% $H_2O$) and the air electrode was exposed to the ambient air. For the electrolysis test, the fuel electrode was fed with 20 mL min$^{-1}$ humidified $H_2$ (3% $H_2O$) and the air electrode was exposed to 200 mL min$^{-1}$ humidified (3% $H_2O$) air. The steam concentration was controlled by a humidification system (Fuel Cell Technologies, Inc.). Faradaic efficiencies were measured based on the ratio of the experimental and theoretical hydrogen generation amounts at fixed current densities. 40 mL min$^{-1}$ 20% $H_2$−80% $N_2$ was fed to the fuel electrode and 60 mL min$^{-1}$ humidified air was fed to the air electrode. Gas chromatography was used to monitor the hydrogen concentration in the fuel electrode, which was used to calculate the amount of actual hydrogen generated. The silver wires were connected with the fuel electrode and air electrode with help of silver paste to collect the current. The current density-voltage curves, as well as the impedance spectra of single cells, were measured using an electrochemical potentiostat (PARSTAT MC 200) at temperatures of 500–650 °C.

**Computational details**. We applied the Vienna ab initio simulation package (VASP)[68,69] to perform density functional theory (DFT) calculations. All calculations were carried out using the projector augmented plane wave (PAW) method[70] with a cut-off energy for the plane-wave basis set of 415 eV. Ba, Co and Fe, and Nb cations with 2+, 3+, and 5+ charge states were chosen to model A-site deficient $Ba_{0.9}Co_{0.7}Fe_{0.2}Nb_{0.1}O_{3-\delta}$ (BCFN) with a cubic structure ($Pm\bar{3}m$). We employed the Hubbard correction[71] with the same effective $U$ value ($U_{eff}$) of 4.0 eV for Co, Fe, and Nb[72,73] and the Perdew-Burke-Ernzerhof (PBE)[74] exchange-correlation functional with the spin-polarized method. Similar to the previous studies[63,75], to investigate the segregation tendency from the bulk to the surface of a B cation on a (001) surface (B = Co, Fe, or Nb), the segregation energy per a B cation was calculated by $E_{segr} = E_{B,bulk}$, where $E_{B,surf}$ and $E_{B,bulk}$ are the total energy for a B dopant atom positioned on the surface and in bulk, respectively.

The cubic crystal structure was optimized without constraints (i.e., volume, shape, or internal structural parameter) and with the spin-polarized and tetrahedron method with Blöchl corrections. For the PBE + U calculations, Brillouin-zone integrations were executed on a grid of ($3 \times 3 \times 3$) **k**-point meshes with the Monkhorst–Pack method[76] to obtain accurate lattice constants. A > 15 Å vacuum space and ($3 \times 3 \times 1$) k-point meshes were employed for surface calculations. Stoichiometric BaBO$_3$ perovskites (B = Co, Fe, or Nb) with a cubic structure ($Pm\bar{3}m$) (4 Ba, 4 B, O 12 atoms)[61] were optimized first to examine the surface energy of 16-layered BO-terminated surfaces (i.e., CoO-, FeO-, and NbO-termination) (Supplementary Table 4 and Supplementary Figs. 12 and 13). Then, $Ba(B_{0.5}B'_{0.5})O_3$ perovskites (8 B, 4 B, 4 B', O 24 atoms) with a cubic structure ($Pm\bar{3}m$) were optimized to investigate the relative segregation energies of Co, Fe, and Nb cation dopants using 16-layered BO- or B'O-terminated surfaces (i.e., CoO-, FeO-, and NbO-termination) (Supplementary Table 5 and Supplementary Figs. 13 and 14). In this study, the (001) surface was chosen for the 2-D slab calculation as the (001) orientation is the most stable[64]. Finally, A-site deficient $Ba_{0.9}Co_{0.7}Fe_{0.2}Nb_{0.1}O_{3-\delta}$ (BCFN) was modeled by constructing $Ba_{0.9}(Co_{0.63}Fe_{0.25}Nb_{0.13})O_{3.0}$ with a cubic structure ($Pm\bar{3}m$) (7 Ba, 5 Co, 2 Fe, 1 Nb, 24 O atoms) (Supplementary Table 6 and Supplementary Figs. 13 and 14) after optimizing $Ba(Co_{0.63}Fe_{0.25}Nb_{0.13})O_{3.0}$ with a cubic structure (8 Ba, 5 Co, 2 Fe, 1 Nb, 24 O atoms). In this study, for the surface calculations of BaBO$_3$ and $Ba(B_{0.5}B'_{0.5})O_3$ perovskites, all of the 16 layers were fully relaxed, while for those of $Ba_{0.9}(Co_{0.63}Fe_{0.25}Nb_{0.13})O_{3.0}$, the bottom four layers were fixed to the bulk properties. The oxygen vacancy formation energy ($E_{OV}$) of BCFN was calculated by $E_{OV} = (E_{def} + E_O) - E_{perf}$, where $E_{def}$ and $E_{perf}$ are the total energy of the slabs with and without an oxygen vacancy, respectively. $E_O$ is the total energy of the triplet gas-phase oxygen atom. It is noted that to avoid unreasonable negative oxygen vacancy formation energies, $E_O$ was applied rather than that of gas-phase oxygen ($O_2$). The oxygen vacancy energy of $Ba(Co_{0.63}Fe_{0.25}Nb_{0.13})O_{3.0}$ was used for comparing with that of $Ba_{0.9}(Co_{0.63}Fe_{0.25}Nb_{0.13})O_{3.0}$ (Supplementary Table 6). As a measure of surface stability of BO-terminated surfaces[62], the surface energies ($E_{surf}$) in the units of J m$^{-2}$ were calculated by $E_{surf} = (E_{slab} - E_{bulk})/2A$, where $E_{slab}$ and $E_{bulk}$ are the total energy of the slab and the bulk unit cell. $A$ is the surface area in the unit of m$^2$. For examining of surface energies, the slabs were fully or partially relaxed as explained in Supplementary Table 4 and Supplementary Table 6. As aforementioned, similar to the previous studies[63,75,77–79], the segregation energy per

a B cation (B = Co, Fe, or Nb) ($E_{segr}$) was calculated as follows. In this study, it is noted that we only considered one dopant's segregation[64].

$$E_{segr} = E_{B,surf} - E_{B,bulk}, \qquad (1)$$

where $E_{B,surf}$ and $E_{B,bulk}$ are the total energy for a B dopant atom (B = Co, Fe, or Nb) positioned on the surface and in bulk, respectively. The smaller the segregation energy, the easier the segregation of the B cation from the bulk to the surface. The bulk diffusion properties[80,81] were calculated using the climbing-image nudged elastic band (CI-NEB) method[82]. The higher oxygen vacancy energy of A-cation deficient $Ba_{0.9}(Co_{0.63}Fe_{0.25}Nb_{0.13})O_{3.0}$ than $Ba(Co_{0.63}Fe_{0.25}Nb_{0.13})O_{3.0}$ (2.95 eV versus 1.87 eV, Supplementary Table 6) leads to a slightly higher diffusion migration barrier (0.60 eV versus 0.58 eV, Supplementary Fig. 12).

## Data availability

The main data generated in this study are provided in the article and Supplementary Information. The source data of Figs. 2a–c, 3a, c–f, 4a–e, and 5b–e, Supplementary Figs. 1–3, 7c, 8a, b, 9a–d, 10, 11a–d, 14, 16k, and 17h are provided as a Source Data file. Other data is available from the corresponding author upon reasonable request. Source data are provided with this paper.

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

## Acknowledgements

Y.C. acknowledges the financial support of the National Natural Science Foundation of China (22179039 and 22005105), the Natural Science Foundation of Guangdong Province (2021A1515010395), the Pearl River Talent Recruitment Program (2019QN01C693, and 2021ZT09L392). K.P. appreciates the support of China Post-doctoral Science Foundation Project (2020M682700). Y.M.C acknowledges the financial support from the Ministry of Science and Technology (MOST Grant No. 110-2221-E-A49-017-MY3), the National Center for High-performance Computing (NCHC), and the Higher Education Sprout Project of the National Yang Ming Chiao Tung University and Ministry of Education (MOE), Taiwan. DFT calculations were partially performed using the resources of the Center for Functional Nanomaterials, which is a U.S. DOE Office of Science Facility, at Brookhaven National Laboratory under Contract No. DE-SC0012704. Y.Z. and M.L. acknowledge the support of CBMM, Hightower Endowed Chair, and Georgia Tech Foundation.

## Author contributions

Y.C. conceived the project. K.P., Y.Z., and K.X. performed the electrochemical testing and preparation and characterization of the materials. H.Z. performed the XRD analysis. Y.D. and B.Z performed the TEM characterization. B.Z. and W.Y. provided the consultation. K.S. and Y.M.C. performed the DFT simulations. Y.M.C, Y.C., and M.L. wrote the manuscript. All authors discussed the results and commented on the manuscript. K.P., Y.Z., and K.X. contributed equally to this work.

## Competing interests

The authors declare no competing interests.
