## [Peer Review File · Nature Communications]

REVIEWER COMMENTS

Reviewer #1 (Remarks to the Author):

Authors in this study have utilised surface segregation of cations to prepare Nb-rich $\text{Ba}_{0.9}\text{Co}_{0.7}\text{Fe}_{0.2}\text{Nb}_{0.1}\text{O}_{3-\delta}$ (BCFN) air electrodes with Nb-deficient BCFN nanoparticle for reversible proton-ceramic electrochemical cells. The performance of the air electrode in fuel-cell mode is comparable to the state-of-the art and the electrode is also showing significant performance in electrolysis mode making it a potential candidate for reversible operation. While surface segregation of cations are known to hinder oxygen surface exchange due to the formation of secondary phases, here the authors have smartly utilised surface structuring as a technique to improve upon electrocatalytic activity. The manuscript may be considered for publication, however; following points has to be noted for revision.

1. In the DFT models, all the surfaces considered are either CoO, FeO or NbO terminated. Whereas for the Ba containing double perovskite in general the surface is known to be BaO terminated or rich in Ba. The segregation of Ba atoms at the surface has been shown to be limiting the efficiency of these double perovskite materials. [Energy Environ. Sci., 2014, 7, 3593–3599; Int. J. Hydrog. Energy, 2014, 39, 20856-20863; Nanoscale, 2019, 11, 21404; ACS applied materials & interfaces, 2019, 11 (28), 25243-25253]

The author should check the stability of BaO terminated surfaces. If the BaO surface is found to be most stable, then the whole discussion should be revised based on the new results.

2. The author should explain why the Nb deficient BCFN nanoparticles are more active.

3. Authors should calculate the oxygen vacancy formation energy and oxygen diffusion coefficient to explain the higher efficiency obtained in this study. [Nanoscale, 2019, 11, 21404; ACS applied materials & interfaces, 2019, 11 (28), 25243-25253]

4. More experimental evidence e.g. Low energy ion scattering (LEIS)) should be used to confirm the Co/Fe segregation. The elemental mapping analysis may be used as a supporting tool. [Chem. Soc. Rev., 2017, 46, 6345]

Reviewer #2 (Remarks to the Author):

This manuscript entitled "Surface Re-structuring of a Perovskite-type Air Electrode for Reversible Protonic Ceramic Electrochemical Cells" deals with re-constructed Nb-rich BCFN electrode covered with Nb-deficient BCFN NPs. The designed cathode was well characterized through appropriate analysis, and the formation mechanism of the designed cathode was elucidated by DFT calculations. This study is expected to contribute to fuel cells and electrolysis researches. However, given the high bar of Nature Communications, there are several issues which need to be addressed prior to publication. Detailed comments are as follows:

1. In Figure 4, Figure 4f is not specified.

2. The captions for (c) and (d) in Figure 5 should be interchanged.

3. Only the gas flow rate applied to the measurement of the symmetric cell is presented. The authors needs to specify the gas flow rate applied to the full cell measurement.

4. In Figure S5, the authors describe that BCFN with less Nb contents shows relatively higher activity. However, although the R_p value is low in BCFN0.05, the E_a value is relatively high. The authors need to clarify this point.

5. The authors need to clarify the advantages of the junction of Nb-rich BCFN electrode and Nb-deficient BCFN NPs compared to single-phase Nb-deficient BCFN.

6. The authors need to present the full cell data measured by applying single-phase Nb-deficient BCFN (ex_BCFN0.05) to the cathode for comparison.

7. In Figure 2d, the authors describe that the formation of multiple BCFN NPs results from

interacting with water. However, in the description of Figure 5f, the explanation about the effect of water on segregation is insufficient. The authors need to clarify this point.

Reviewer #3 (Remarks to the Author):

Recently, reversible protonic ceramic electrochemical cells (PCECs) have emerged as a new and efficient energy storage and conversion device. This work focused on addressing the tough challenges of the air electrode: the poor performance and insufficient stability of the air electrode. Overall, the experiments have been designed and performed nicely. The excellent cell performance (peak power density in fuel cell mode, current density at an electrolysis voltage) and demonstration of stability for over 300 hours warrant publication. The manuscript is also well organized. I believe that the results reported in this manuscript are of big interest to the broad readership of Nat Commun. The manuscript can be recommended for publication in Nat Commun after some minor revisions. The detailed comments/suggestions are shown below:

1. The mass transfer is very important, especially when the cell is in the electrolysis mode. What is the pore structure of the cells fabricated for this study? The authors need to provide such information, at least the SEM image with more details of the fuel electrode and the air electrode.
2. Insufficient information about the cell testing, such as the active surface area of the cells, flow rates of gases, is provided. That information is very important when evaluating the cell Faradaic efficiency. In addition, more details about the FE measurement should be added since a small cell area would lead to large uncertainty.
3. The most interesting point of this work is the surface re-structuring of the air electrode. However, such information is not provided in detail. What temperature did those nanoparticles start to exsolve? Is the exsolution process reversible?

Point-to-point Responses to Reviewers

First of all, we very much appreciate the reviewers' time and efforts spent on reviewing our manuscript. The Reviewers' thoughtful comments and suggestions have significantly improved the quality and clarity of our manuscript. We very much appreciate the reviewers' comments on our manuscript. We have addressed all the comments and revised the manuscript accordingly. Our point-to-point responses are presented below.

Reviewer(s)' Comments to Author:

Reviewer #1:

Authors in this study have utilised surface segregation of cations to prepare Nb-rich $\text{Ba}_{0.9}\text{Co}_{0.7}\text{Fe}_{0.2}\text{Nb}_{0.1}\text{O}_{3-\delta}$ (BCFN) air electrodes with Nb-deficient BCFN nanoparticle for reversible proton-ceramic electrochemical cells. The performance of the air electrode in fuel-cell mode is comparable to the state-of-the art and the electrode is also showing significant performance in electrolysis mode making it a potential candidate for reversible operation. While surface segregation of cations are known to hinder oxygen surface exchange due to the formation of secondary phases, here the authors have smartly utilised surface structuring as a technique to improve upon electrocatalytic activity. The manuscript may be considered for publication, however; following points has to be noted for revision.

Reply to general comments:

We highly appreciate the positive comments from Reviewer #1.

Comment #1of Reviewer 1

In the DFT models, all the surfaces considered are either CoO, FeO or NbO terminated. Whereas for the Ba containing double perovskite in general the surface is known to be BaO terminated or rich in Ba. The segregation of Ba atoms at the surface has been shown to be limiting the efficiency of these double perovskite materials. [Energy Environ. Sci., 2014, 7, 3593–3599; Int. J. Hydrog. Energy, 2014, 39, 20856-20863; Nanoscale, 2019, 11, 21404; ACS applied materials & interfaces, 2019,

11 (28), 25243-25253]

The author should check the stability of BaO terminated surfaces. If the BaO surface is found to be most stable, then the whole discussion should be revised based on the new results.

Reply to C1: We thank the reviewer for this valuable comment.

Due to the high complexity of A-cation deficient $\text{Ba}_{0.9}(\text{Co}_{0.63}\text{Fe}_{0.25}\text{Nb}_{0.13})\text{O}_{3.0}$, as clarified in the main text and the supplementary section, stoichiometric BaBO_3 (B = Co, Fe, or Nb) was first examined. To follow the reviewer's good suggestion, we have calculated the surface stability of BaO-terminated surfaces (see updated **Supplementary Table S3**) of BaBO_3 (B = Co, Fe, or Nb), leading to the surface energy of BaO termination (0.45, 0.85, and 0.94 J/m^2 , respectively, versus CoO: 0.51 J/m^2 , FeO: 0.82 J/m^2 , and NbO: 0.93 J/m^2). This shows that the BaO-terminated surface's stability might depend on the interaction of BaO with B cations. Also, we examined the BaO-terminated surface's stability of different surface models using $\text{Ba}_{0.9}(\text{Co}_{0.63}\text{Fe}_{0.25}\text{Nb}_{0.13})\text{O}_{3.0}(001)$ (**Supplementary Table S5**). Similar to the results using $\text{BaBO}_3(001)$ models, we observed that the surface stability is in the order of CoO (0.78 J/m^2), BaO (0.93 J/m^2), and CoFeNbO (1.03 J/m^2). This may qualitatively support the validity of our approach, examining the segregation of Co, Fe, and Nb after examining the surface stability of $\text{BaBO}_3(001)$ (B = Co, Fe, or Nb). To fully follow the reviewer's important comment, we have revised the supporting information (**Supplementary Tables S3 and S5**).

Revision: We have made the following revisions in the main text and revised the Supplementary Table S3 and S5 accordingly.

“We also observed that the surface energy of their BaO-terminated surface depends on B cations (**Supplementary Table S3**).” and “We also assumed that the segregation of Ba atoms may be less important than B cations as the surface energies of BaO-terminated surfaces rely on B cations (*i.e.*, BaO-CoO-BaO-CoFeNbO- (0.95 J/m^2) and BaO-CoFeNbO-BaO-CoO- (0.91 J/m^2)).”

Supplementary Table S3. Optimized lattice parameters (a_0) of $BaBO_3$ (B = Co, Fe, or Nb) and the surface energies of BaO- and BO-terminated $BaBO_3(001)$.

	bulk	slab	
	a_0 (Å)	surface energy (J/m ²)	termination
BaCoO ₃	3.9562	0.45	BaO
		0.51	CoO
BaFeO ₃	3.9761	0.81	BaO
		0.82	FeO
BaNbO ₃	4.1208	0.94	BaO
		0.93	NbO
Surface stability of BO termination			CoO > FeO > NbO

Note: In this comparison, all of layers were fully relaxed without any restriction. As shown in the summary of BaO-terminated surfaces, the stability may be dependent on B cation types.

Comment #2 of Reviewer 1

The author should explain why the Nb deficient BCFN nanoparticles are more active.

Response: We very much appreciate this comment. We apologize for the unclear statement. Nb-deficient BCFN (for example, $Ba_{0.9}Co_{0.7}Fe_{0.2}Nb_{0.05}O_3$) shows higher activity than BCFN with no deficiency ($Ba_{0.9}Co_{0.7}Fe_{0.2}Nb_{0.1}O_3$). It is reported that less reducible cations such as Zr, Ta, or Nb are expected to enhance the durability of perovskite materials (*N. Tsvetkov, Q. Lu, L. Sun, E. J. Crumlin and B. Yildiz, Nat Mater, 2016, 15, 1010-1016; M. Li, M. Zhao, F. Li, W. Zhou, V. K. Peterson, X. Xu, Z. Shao, I. Gentle and Z. Zhu, Nat Commun, 2017, 8, 13990.*). An Nb-deficient BCFN material on the surface is expected to have a higher concentration of oxygen vacancy than BCFN in the bulk (without an Nb deficiency), as confirmed by thermogravimetric analyses of BCFN0.1 and BCFN0.05 (**Figure R1**). The valence changes of Co or Fe would provide more oxygen vacancies at high temperatures in the oxygen reduction or evolution reactions, which is likely the reason for the higher activity of Nb-deficient BCFN nanoparticles.

Figure R1 Thermalgravimetric analyses of BCFN0.05 and BCFN0.1

Revision: We have added related discussion to the revised manuscript.

In the main text: “The results indicated that BCFN with less Nb shows a relatively higher activity, acquired from both symmetrical cells and full cells (Supplementary Fig. S8, Fig. S9, Fig. S10, and Fig. S11).”

In the Supporting Information, “Nb-deficient BCFN (for example, $\text{Ba}_{0.9}\text{Co}_{0.7}\text{Fe}_{0.2}\text{Nb}_{0.05}\text{O}_3$) shows higher activity than BCFN with no deficiency ($\text{Ba}_{0.9}\text{Co}_{0.7}\text{Fe}_{0.2}\text{Nb}_{0.1}\text{O}_3$). It is reported that less reducible cations such as Zr, Ta, or Nb are expected to enhance the durability of perovskite materials (N. Tsvetkov, Q. Lu, L. Sun, E. J. Crumlin and B. Yildiz, *Nat Mater*, 2016, 15, 1010-1016; M. Li, M. Zhao, F. Li, W. Zhou, V. K. Peterson, X. Xu, Z. Shao, I. Gentle and Z. Zhu, *Nat Commun*, 2017, 8, 13990.). An Nb-deficient BCFN material on the surface is expected to have a higher concentration of oxygen vacancy than BCFN in the bulk (without an Nb deficiency), as confirmed by thermogravimetric analyses of BCFN0.1 and BCFN0.05 (Figure S6). The valence changes of Co or Fe would provide more oxygen vacancies (more weight loss) at high temperatures in the oxygen reduction or evolution reactions,

which is likely the reasons for the higher activity of Nb-deficient BCFN nanoparticles.”

Comment #3 of Reviewer 1

Authors should calculate the oxygen vacancy formation energy and oxygen diffusion coefficient to explain the higher efficiency obtained in this study. [Nanoscale, 2019, 11, 21404; ACS applied materials & interfaces, 2019, 11 (28), 25243-25253]

Response to C3: We genuinely appreciate the reviewer’s comments. Our initial manuscript added an averaged oxygen vacancy formation energy of A-cation deficient $\text{Ba}_{0.9}(\text{Co}_{0.63}\text{Fe}_{0.25}\text{Nb}_{0.13})\text{O}_{3.0}$. (**Supplementary Table S5**). Thanks to the reviewer’s good suggestion, we could compare the oxygen vacancy formation energies of A-cation deficient $\text{Ba}_{0.9}(\text{Co}_{0.63}\text{Fe}_{0.25}\text{Nb}_{0.13})\text{O}_{3.0}$ and $\text{Ba}(\text{Co}_{0.63}\text{Fe}_{0.25}\text{Nb}_{0.13})\text{O}_{3.0}$ as explained in the computational method and Supplementary Table S5 revised (2.95 eV versus 1.87 eV, respectively). Simulating diffusion coefficients is beyond the scope of this study, and it could be added to our potential publications in the near future. However, since an understanding of bulk diffusion characteristics is important, we carried out DFT-based migration energy calculations by using the CI-NEB method as they are correlated with ionic conductivities ($D = D_0 \cdot \exp(-E_a/RT)$). Diffusion barrier E_a is further described by $E_a = E_m + E_{OV}$, in which E_m and E_{OV} are the migration energy of oxygen vacancies and the oxygen vacancy formation energy, respectively (Reference: *PNAS*, **2006**, 103(10), 3518-3521). As a good comparison, we calculated the migration energy of A-cation deficient $\text{Ba}_{0.9}(\text{Co}_{0.63}\text{Fe}_{0.25}\text{Nb}_{0.13})\text{O}_{3.0}$ and $\text{Ba}(\text{Co}_{0.63}\text{Fe}_{0.25}\text{Nb}_{0.13})\text{O}_{3.0}$ along the [001] direction, resulting in a migration energy of 0.60 eV and 0.58 eV, respectively. We observed that the higher oxygen vacancy energy of A-cation deficient $\text{Ba}_{0.9}(\text{Co}_{0.63}\text{Fe}_{0.25}\text{Nb}_{0.13})\text{O}_{3.0}$ than $\text{Ba}(\text{Co}_{0.63}\text{Fe}_{0.25}\text{Nb}_{0.13})\text{O}_{3.0}$ (2.95 eV versus 1.87 eV, **Supplementary Table S5**) leads to a slightly higher diffusion migration barrier (0.60 eV versus 0.58 eV, **Supplementary Figure S12**). Accordingly, Supplementary Figure S12 and Table S5 have been revised. The two suggested references (References: 1. *Nanoscale*, **2019**, 11, 21404 and 2. *ACS Applied Materials & Interfaces*, **2019**, 11 (28), 25243-25253) have

been cited in the computational methods. In particular, to make it more informative, we have revised the supporting information.

Revision: We have made revisions in the computational method in the Supporting Information.

“The bulk diffusion properties^{17,18} are examined by using the climbing-image nudged elastic band (CI-NEB) method¹⁹. The higher oxygen vacancy energy of A-cation deficient $\text{Ba}_{0.9}(\text{Co}_{0.63}\text{Fe}_{0.25}\text{Nb}_{0.13})\text{O}_{3.0}$ than $\text{Ba}(\text{Co}_{0.63}\text{Fe}_{0.25}\text{Nb}_{0.13})\text{O}_{3.0}$ (2.95 eV versus 1.87 eV, **Supplementary Table S5**) leads to a slightly higher diffusion migration barrier (0.60 eV versus 0.58 eV, **Supplementary Figure S12**).”

Supplementary Table S5. Optimized lattice constant (a_0) and oxygen vacancy formation energies (E_{OV}) of $\text{Ba}_{0.9}(\text{Co}_{0.63}\text{Fe}_{0.25}\text{Nb}_{0.13})\text{O}_{3.0}$ (BCFN) and surface energies of $\text{Ba}_{0.9}(\text{Co}_{0.63}\text{Fe}_{0.25}\text{Nb}_{0.13})\text{O}_{3.0}(001)$ (BCFN(001)).

	bulk		slab	
	a_0 (Å)	E_{OV} (eV) ¹	termination	surface energy (J/m ²) ²
$\text{Ba}_{0.9}(\text{Co}_{0.63}\text{Fe}_{0.25}\text{Nb}_{0.13})\text{O}_{3.0}$ (7 Ba, 5 Co, 2 Fe, 1 Nb, 24 O)	3.9768	2.95	CoO	0.78
			CoFeNbO	1.03
			BaO	0.93

Note 1: The oxygen vacancy formation energy was averaged after calculating at 24 oxygen vacancy positions. An oxygen vacancy formation energy of $\text{Ba}(\text{Co}_{0.63}\text{Fe}_{0.25}\text{Nb}_{0.13})\text{O}_{3.0}$ with a cubic structure (8 Ba, 5 Co, 2 Fe, 1 Nb, 24 O atoms) calculated is 1.87 eV, indicating ~37% more difficult to generate oxygen vacancies.

Note 2: It is an averaged surface energy of two BaO-terminated models (1. BaO-CoFeNbO-BaO-CoO- (0.91 J/m²) and 2. BaO-CoO-BaO-CoFeNbO- (0.95 J/m²)). The surface models were prepared with 16 layers, and the bottom eight layers were fixed at the bulk parameters.

Supplementary Table S6. Segregation energies of $\text{Ba}_{0.9}(\text{Co}_{0.63}\text{Fe}_{0.25}\text{Nb}_{0.13})\text{O}_{3.0}(001)$.

surface configuration ¹	segregation energy (eV) ²				remark
	Co	Fe ₁	Fe ₂	Nb	
CoO	0.39	0.71	2.33	1.18	before swapping Co (surface) and Nb (bulk)

CoNbO	!	0.09	-0.04	!	before swapping Co (surface) and Nb (bulk)
-------	---	------	-------	---	--

1. The energy difference of CoNbO (after swapping Nb and Co atoms) and CoO (before switching Nb and Co atoms) is 0.17 J/m^2 , indicating that Nb makes the surface less stable than the CoO-terminated surface.
2. The segregation energy of Fe becomes smaller after an Nb atom segregates to the surface. It explains that Nb that locally segregates to the surface augments the segregation of Fe.

Comment #4 of Reviewer 1

More experimental evidence e.g. Low energy ion scattering (LEIS)) should be used to confirm the Co/Fe segregation. The elemental mapping analysis may be used as a supporting tool. [Chem. Soc. Rev., 2017, 46, 6345]

Response to C4: We very much appreciate this suggestion from Reviewer #1. We agree with reviewer #1 that low energy ion scattering (LEIS) spectroscopy is a powerful technique to probe the composition and structure of the outermost surface. The probing depth of LEIS is typically one or two atomic layers of the surface due to the relatively high probability of ionic neutralization at the surface, making LEIS one of the most sensitive surface analysis methods (*Y. Li, W. Zhang, Y. Zheng, J. Chen, B. Yu, Y. Chen, and M. Liu, Chemical Society Reviews, 2017, 46, 6345-6378.*). However, it would need an extremely amount of effort to make a high-quality film for this technique in order to get a meaningful result. In addition, this technique is not always available to the researchers. Therefore, we request that the reviewer would consider our other evidence (electron energy loss spectroscopy, and elemental mapping) to confirm the segregation of Co/Fe, as shown in Figure R5 below. We are eager to find any opportunity to perform a LEIS experiment, which could be included in our future publications.

Figure R2 (a) and (b) A STEM image of a BCFN grain covered with nanoparticles; and **(c)** an electron energy-loss spectroscopy (EELS) profile along the white line shown in (a).

Shown in **Figures R2a** and **R2b** are the STEM images of a BCFN grain covered with nanoparticles. Shown in **Figure R2c** is the electron energy-loss spectroscopy (EELS) profile of Co and Fe along the scanning line shown in Figure R2b. Slightly higher contents of Co or Fe are observed in the EELS profile (Figure R2c), suggesting a possible Co/Fe segregation. In addition, we performed elemental mappings of Ba, Co, Fe, and Nb of a BCFN grain covered with a nanoparticle, as shown in **Figures R3** and **R4**. It is confirmed that there is a Co/Fe segregation within the nanoparticle. In other words, Nb deficiency in the nanoparticle is confirmed.

Figure R3 An elemental mapping of BCFN grain covered with a nanoparticle (a).
Mapping of Ba (b), Co (c), Fe (d), Nb (e), and all (f)

Figure R4 An elemental mapping of a BCFN grain covered with a nanoparticle (a).

Mapping of Ba (b), Co (c), Fe (d), Nb (e), and all (f).

Revision: We have added **Figures S6** and **S7**, and related discussion in the revised Supporting Information.

Supplementary Figure S6 An elemental mapping of BCFN grain covered with a nanoparticle (a). Mapping of Ba (b), Co (c), Fe (d), Nb (e), and all (f).

We performed elemental mappings of Ba, Co, Fe, and Nb of a BCFN grain (after test) covered with a nanoparticle, as shown in **Figures S6**. It is confirmed that there is a Co/Fe segregation within the nanoparticle. In other words, Nb deficiency in the nanoparticle is confirmed.

Supplementary Figure S7 (a) and (b) A STEM image of a BCFN grain covered with nanoparticles; and (c) an electron energy-loss spectroscopy (EELS) profile along the white line shown in (a).

Shown in **Figures S7a** and **S7b** are the STEM images of a BCFN grain covered with nanoparticles. Shown in **Figure S7c** is the electron energy-loss spectroscopy (EELS) profile of Co and Fe along the scanning line shown in **Figure S7b**. Slightly higher contents of Co or Fe are observed in the EELS profile (**Figure S7c**), suggesting a possible Co/Fe segregation.

Reviewer #2:

This manuscript entitled “Surface Re-structuring of a Perovskite-type Air Electrode for Reversible Protonic Ceramic Electrochemical Cells” deals with re-constructed Nb-rich BCFN electrode covered with Nb-deficient BCFN NPs. The designed cathode was well characterized through appropriate analysis, and the formation mechanism of the designed cathode was elucidated by DFT calculations. This study is expected to contribute to fuel cells and electrolysis researches. However, given the high bar of Nature Communications, there are several issues which need to be

addressed prior to publication. Detailed comments are as follows:

Reply to general comments:

We highly appreciate the comment and recommendation from Reviewer #2.

Comment #1 of Reviewer 2

In Figure 4, Figure 4f is not specified.

Response to C1: We thank the reviewer very much for the careful reading. We apologize for this typo. The figure we intended to point to is Fig. 4e.

Revision: We have corrected this typo in the revised manuscript.

“It is also consistent with the performance of cells at 1 A cm^{-2} current density and at 600°C when the water concentration achieved at 30vol.%, 40vol.%, and 50vol.%, faradaic efficiency increased obviously from 85% to 92% (as shown in the right figure of Fig. 4e).

”

Comment #2 of Reviewer 2

The captions for (c) and (d) in Figure 5 should be interchanged.

Response to C2: We thank the reviewer again for the careful reading. We have interchanged the captions for (c) and (d) in the revised manuscript.

Revision: We have interchanged the captions for (c) and (d) in the revised manuscript.

“**Figure 5. Formation of the high performance air electrode of BCFN covered with fine BCFN nanoparticles with less Nb. (a)** TEM image and elemental (Ba, Co Fe, and Nb) mapping of a BCFN grain after electrochemical testing at 650°C ; **(b)** Elemental profile along the scanning line shown in the TEM image; **(c)** Comparison of computed segregation energy (J/m^2) for

stoichiometric $\text{Ba}(\text{B}_{0.5}\text{B}'_{0.5})\text{O}_3$ (B,B' = Co, Fe, or Nb); (d) Illustration of CoO-terminated BCFN(001) surface models representing segregation of Nb from the bulk to the surface; (e) Computed segregation energy (eV) for three different dopant cations of $\text{Ba}_{0.9}(\text{Co}_{0.63}\text{Fe}_{0.25}\text{Nb}_{0.13})\text{O}_{3.0}$; and (f) A schematic illustration of the formation of Nb-deficient BCFN nanoparticles on the BCFN air electrode.

”

Comment #3 of Reviewer 2

Only the gas flow rate applied to the measurement of the symmetric cell is presented. The authors needs to specify the gas flow rate applied to the full cell measurement.

Response to C3: We appreciate the reviewer very much for this comment. For the fuel cell test, the fuel electrode was exposed to 20 mL min^{-1} humidified H_2 (3% H_2O) and the air electrode was exposed to ambient air. For the electrolysis test, the fuel electrode was fed with 20 mL min^{-1} humidified H_2 (3% H_2O) and the air electrode was exposed to 200 mL min^{-1} humidified (3% H_2O) air. The steam concentration was controlled by a humidification system (Fuel Cell Technologies, Inc.). Faradic efficiencies (η_F) were measured based on the ratio of the experimental and theoretical hydrogen generation amounts at fixed current densities. 40 mL min^{-1} 20% H_2 -80% N_2 was fed to the fuel electrode and 60 mL min^{-1} humidified air was fed to the air electrode. Gas chromatography was used to monitor the hydrogen concentration in the fuel electrode, which was used to calculate the amount of actual hydrogen generated.

Revision: We have added those details in the experimental section in the revised manuscript.

“For the fuel cell (small button cell) test, the fuel electrode was exposed to 20 mL min^{-1} humidified H_2 (3% H_2O) and the air electrode was exposed to ambient air. For the electrolysis test, the fuel electrode was fed with 20 mL min^{-1} humidified H_2 (3% H_2O) and the air electrode was exposed to 200 mL min^{-1} humidified (3% H_2O) air.

The steam concentration was controlled by a humidification system (Fuel Cell Technologies, Inc.). Faradic efficiencies (η_F) were measured based on the ratio of the experimental and theoretical hydrogen generation amounts at fixed current densities. 40 mL min⁻¹ 20% H₂-80% N₂ was fed to the fuel electrode and 60 mL min⁻¹ humidified air was fed to the air electrode. Gas chromatography was used to monitor the hydrogen concentration in the fuel electrode, which was used to calculate the amount of actual hydrogen generated.”

Comment #4 of Reviewer 2

In Figure S5, the authors describe that BCFN with less Nb contents shows relatively higher activity. However, although the R_p value is low in BCFN0.05, the E_a value is relatively high. The authors need to clarify this point.

Response: We appreciate Reviewer#4 for the careful reading. The reviewer has corrected noted that Nb-deficient BCFN showed a higher E_a. Actually, we have prepared a considerable number of symmetrical cells with those electrodes and measured the R_p of each electrode. Similar values of R_p and E_a were obtained, indicating that our data is repeatable. The relatively higher E_a value for Nb deficient BCFN might suggest that a higher reaction barrier has to be overcome. It is shown in **Figure S5a** that, R_p values of BCFN0.05, BCFN0.1, and BCFN0.2 are similar at a low temperature of 550°C. The difference became more pronounced as the operating temperature increased: Nb-deficient BCFN showed a lower R_p value, which is likely caused by the more oxygen vacancy formed at higher temperatures. Nb deficient BCFN is expected to have more oxygen vacancy at higher temperatures, as evidenced by the TGA results (**Figure R2**).

Revision: We have added related discussion to the revised Supporting Information.

“A relatively higher E_a value for Nb deficient BCFN was observed, suggesting that there is a higher reaction barrier has to be overcome in the oxygen reaction process. It is shown that R_p values of BCFN0.05, BCFN0.1, and BCFN0.2 were similar at a low

temperature of 550 °C. The difference became more pronounced as the operating temperature increased: Nb-deficient BCFN showed a lower R_p value, which is likely caused by the more oxygen vacancy formed at higher temperatures. Nb deficient BCFN is expected to have more oxygen vacancy at higher temperatures.”

Comment #5 of Reviewer 2

The authors need to clarify the advantages of the junction of Nb-rich BCFN electrode and Nb-deficient BCFN NPs compared to single-phase Nb-deficient BCFN.

Response: We thank the reviewer very much for this suggestion. The electrode system (Nb-rich BCFN electrode covered by Nb-deficient BCFN NPs) showed a higher reaction activity compared to the single-phase Nb-deficient BCFN, as shown in the manuscript. In order to understand the effect of Nb content in the BCFN system on the activity and stability of $Ba_{0.9}Co_{0.7}Fe_{0.2}Nb_x$ ($x=0.025, 0.05, \text{ and } 0.1$, denoted as BCFN $_x$), we prepared BZCYYb electrolyte based symmetrical cells with BCFN $_x$ electrode. Shown in **Figure R5** are the impedance spectra of BCFN $_x$ electrode and the R_p evolution as a function of testing time. It is shown that BCFN $_x$ with less Nb showed a higher activity but worse durability, while BCFN with more Nb showed a lower activity but better durability. To combine those advantages, it is reasonably expected that our junction of Nb-rich BCFN electrode and Nb-deficient BCFN NPs showed high activity and good durability.

Figure R5 Effect of Nb content in $\text{Ba}_{0.9}\text{Co}_{0.7}\text{Fe}_{0.2}\text{Nb}_x$ ($x=0.025, 0.05, \text{ and } 0.1$, denoted as BCFNx) on the electrode activity and durability. Electrochemical impedance spectra of BCFNx at different testing time and $600\text{ }^\circ\text{C}$: $x=0.025$ (a); $x=0.05$ (b); and $x=0.1$ (c); (d) Stability of R_p values of BCFNx electrodes in 100h.

It has been reported that the *in-situ* formation of the Nb-deficient BCFN nanoparticles on BCFN electrode is expected to have better structural stability since the risk of agglomeration is greatly reduced, thus enhancing the durability and thermal stability of the electrode. Such enhanced stability is also observed in other reports [W. Kobsiriphat, B. D. Madsen, Y. Wang, M. Shah, L. D. Marks and S. A. Barnett, *Journal of The Electrochemical Society*, 2010, 157, B279.; D. Neagu, T.-S. Oh, D. N. Miller, H. Ménard, S. M. Bukhari, S. R. Gamble, R. J. Gorte, J. M. Vohs and J. T. S. Irvine, *Nature Communications*, 2015, 6, 8120.; R. Shiozaki, A. G. Andersen, T. Hayakawa, S. Hamakawa, K. Suzuki, M. Shimizu and K. Takehira, *Journal of the Chemical Society, Faraday Transactions*, 1997, 93, 3235-3242.]. In addition, the nano-particles maintained a crystallographic coherence with the host perovskite lattice and consequently may experience a lattice strain [D. Neagu, T.-S. Oh, D. N. Miller, H. Ménard, S. M. Bukhari, S. R. Gamble, R. J. Gorte, J. M. Vohs and J. T. S. Irvine, *Nature Communications*, 2015, 6, 8120.; D. Neagu, G. Tsekouras, D. N. Miller, H.

Ménard and J. T. S. Irvine, *Nature Chemistry*, 2013, 5, 916-923.]. Such confined and seemingly anchored particles are expected to exhibit considerably different physical and chemical properties as compared with the unconstrained particles, introduced by the traditional solution infiltration process [J. T. S. Irvine, D. Neagu, M. C. Verbraeken, C. Chatzichristodoulou, C. Graves and M. B. Mogensen, *Nature Energy*, 2016, 1, 15014; S. P. Jiang, *Materials Science and Engineering: A*, 2006, 418, 199-210.].

Revision: We have added related discussions in the revised manuscript or Supporting Information.

In the main text: “The *in-situ* formed Nb-deficient BCFN nanoparticles on BCFN electrodes is expected to have better structural stability since the risk of agglomeration is greatly reduced, thus enhancing the durability and thermal stability of the electrode 59-62.”

In the Supporting information: “The electrode system (Nb-rich BCFN electrode covered by Nb-deficient BCFN NPs) showed a higher reaction activity compared to the single-phase Nb-deficient BCFN, as shown in the manuscript. In order to understand the effect of Nb content in BCFN system on the activity and stability of $\text{Ba}_{0.9}\text{Co}_{0.7}\text{Fe}_{0.2}\text{Nb}_x$ ($x=0.025, 0.05, \text{ and } 0.1$, denoted as BCFN $_x$), we prepared BZCYYb electrolyte based symmetrical cells with BCFN $_x$ electrode. Shown in **Figure S9** are the impedance spectra of BCFN $_x$ and the evolution of R_p as a function of temperatures. It is shown that BCFN $_x$ with less Nb showed a higher activity but worse durability, while BCFN with more Nb showed a lower activity but better durability. To combine those advantages, it is reasonably expected that our junction of Nb-rich BCFN electrode and Nb-deficient BCFN NPs showed higher activity and better durability.

Supplementary Figure S9 Effect of Nb content in $\text{Ba}_{0.9}\text{Co}_{0.7}\text{Fe}_{0.2}\text{Nb}_x$ ($x=0.025, 0.05,$ and 0.1 , denoted as BCFN x) on the electrode activity and durability. Electrochemical impedance spectra of BCFN x at different times and $600\text{ }^\circ\text{C}$: $x=0.025$ (a); $x=0.05$ (b); and $x=0.1$ (c); (d) R_p changes as a function of testing time in 100h.

It has been reported that the in-situ formation of the Nb deficient BCFN nanoparticles on BCFN electrodes is expected to have better structural stability since the risk of agglomeration is greatly reduced, thus enhancing the durability and thermal stability of the electrode. Such enhanced stability is also observed in other reports^{20,21}. In addition, the nano-particles maintained a crystallographic coherence with the host perovskite lattice and consequently may experience lattice strain^{21,22}. Such confined and seemingly anchored particles are expected to exhibit considerably different physical and chemical properties as compared with the unconstrained particles, introduced by the traditional solution infiltration process^{23,24}.

Comment #6 of Reviewer 2

The authors need to present the full cell data measured by applying single-phase Nb-deficient BCFN (ex_BCFN0.05) to the cathode for comparison.

Response: We appreciate the reviewer very much for this suggestion. We have

prepared the full cell when an Nb-deficient BCFN (BCFN0.05) was applied as the cathode. The results are shown in **Figure R6 a-d**. It is found that the full cell with BCFN0.05 air electrode and BCFN (in the main text) showed similar peak power densities: 1.69 W cm^{-2} for BCFN0.05 vs 1.70 W cm^{-2} for BCFN in the main text, which is reasonable since the full cell performance can be affected by factors such as anode, electrolyte and cathode, and especially the electrode/electrolyte interfaces. However, full cells with BCFN0.05 electrode showed much worse stability, in modes of fuel cells and electrolysis cells (**Figures R6 c and d**).

Figure R6 Performance of a full cell with a BCFN0.05 air electrode. **(a)** Typical IVP curves in a fuel cell mode measured at 500-650 °C; **(b)** Typical IV curve in an electrolysis mode measured at 500-600 °C; **(c)** A short-term durability test of a fuel cell tested at 600 °C and a current density of $+0.5 \text{ A cm}^{-2}$. and **(d)** A short-term durability test of an electrolysis cell tested at 550 °C and a current density of -0.5 A cm^{-2}

Revision: We have added **Supplementary Figure S11** in the revised Supporting Information.

Supplementary Figure S11 Performance of a full cell with a BCFN0.05 air electrode.

(a) Typical IVP curves in a fuel cell mode measured at 500-650 °C; (b) Typical IV curve in an electrolysis mode measured at 500-600 °C; (c) A short-term durability test of a fuel cell tested at 600 °C and a current density of +0.5 A cm⁻². and (d) A short-term durability test of an electrolysis cell tested at 550 °C and a current density of -0.5 A cm⁻²

Comment #7 of Reviewer 2

In Figure 2d, the authors describe that the formation of multiple BCFN NPs results from interacting with water. However, in the description of Figure 5f, the explanation about the effect of water on segregation is insufficient. The authors need to clarify this point.

Response to C7: We very much appreciate the reviewer for this comment. We have performed additional experiments to clarify the interaction of water with BCFN,

including the effect of different treating temperatures, and different treating times at 600°C on the exsolution. We also evaluate if the voltage/current or the chemical potential of steam is the main reason for the exsolution.

First, we operated the BCFN air electrode at 600 °C in the air with 3% H₂O for different durations (5-40 hours) at OCV conditions to disclose the relationship between the amount of exsolved nanoparticles and operating time (**Figure R7**). As shown, a considerable number of nanoparticles (with a population of ~20 μm⁻²) have been exsolved after 5 hours of treatment. The precipitated nanoparticles significantly increase as a function of treating time (in the following 35h). However, after 10h, the NP population tends to be stable.

Figure R7. Detailed SEM images of BCFN air electrode after operated under 3% humidified air at 600 °C for 5 hours (a) & (d), 10 hours (b) & (e), 20 hours (c) & (f),

30 hours (g) & (i), 40 hours (h) & (j), respectively. (k) Time dependence of exsolved nanoparticles populations of BCFN air electrode operated at 3% humidified H₂O at 600 °C.

Figure R8 Detailed SEM images of BCFN air electrode after operated under 3% humidified air for 10 hours at (a) room temperature, (b) 400 °C, (c) 450 °C, (d) 500 °C, (e) 550 °C and (f) 600 °C, respectively. (g) Typical SEM images of BCFN air electrode after operated with 3% H₂O humidified air at 600 °C for 10 hours and followed operated with dry air at 600 °C for 10 hours. Temperature dependent exsolved nanoparticles population of BCFN air electrode operated at 3% H₂O humidified air for 10 hours. (h) populations of the exsolved nanoparticles with diameter of 50-100 nm (blue) and over 100 nm (purple) when BCFN electrode was operated at 3% humidified H₂O for 10 hours at 400-600 °C.

In order to investigate the specific temperature that nanoparticles start to exsolve from the skeleton, we operated the BCFN air electrode at different temperatures (400-600 °C) with 3% humidified air for 10 hours at OCV condition (**Figure R8**). It is revealed that BCFN approximately started to exsolve at 400 °C. Similarly, we roughly

estimated the population of the exsolved nanoparticles (NP) and found that the NP population (over 100nm) increased significantly as temperature increased.

Figure R9. Typical SEM images of BCFN air electrode after operated at OCV condition with wet (3% H₂O) hydrogen as fuel and dry air as oxidant (a) & (d), a current density of -0.5 A cm⁻² with wet (3% H₂O) hydrogen as fuel and dry air as oxidant (b) & (e), and a current density of +0.5 A cm⁻² with wet (3% H₂O) hydrogen as fuel and dry air as oxidant (c) & (f) at 600 °C for 10 hours; Typical SEM images of BCFN air electrode after operated at OCV condition with wet (3% H₂O) hydrogen as fuel and wet (3% H₂O) air as oxidant (g) & (j), a current density of -0.5 A cm⁻² with wet (3% H₂O) hydrogen as fuel and wet (3% H₂O) air as oxidant (h) & (k) and a current density of +0.5 A cm⁻² with wet (3% H₂O) hydrogen as fuel and wet (3% H₂O) air as oxidant (i) & (l) at 600 °C for 10 hours.

To evaluate if the voltage/current or the chemical potential of steam is the main reason

for the exsolution, we treated the air electrode under the following conditions. First, we operated cells with BCFN air electrode at OCV condition when the air electrode is exposed to dry air at 600 °C for 10 hours at current densities of 0, -0.5 and +0.5 Acm⁻². It is shown that very few nanoparticles are exsolved (**Figure R9a-f**). In contrast, when the air electrode is exposed to wet air (3% H₂O) at current densities of 0, -0.5, and +0.5 Acm⁻², a great number of nanoparticles were observed on the skeleton (Figure R9 g-l), indicating that steam can lead to the exsolution. Therefore, it is reasonable to conclude that the current flowing may not be the factor required to induce the exsolution. The steam is most likely the main reason for the exsolution.

Revision: We have added **Supplementary Figures S16, S17 and S18**, and related discussions in the revised Supporting Information.

First, we operated the BCFN air electrode at 600 °C in the air with 3% H₂O for different durations (5-40 hours) at OCV conditions to disclose the relationship between the amount of exsolved nanoparticles and operating time (**Figure S16**). As shown, a considerable number of nanoparticles (with a population of ~20 μm⁻²) have been exsolved after 5 hours of treatment. The precipitated nanoparticles significantly increase as a function of treating time (in the following 35h). However, after 10h, the NP population tends to be stable.

Supplementary Figure S16. Detailed SEM images of BCFN air electrode after operated under 3% humidified air at 600 °C for 5 hours (a) & (d), 10 hours (b) & (e), 20 hours (c) & (f), 30 hours (g) & (i), 40 hours (h) & (j), respectively. (k) Time dependence of exsolved nanoparticles populations of BCFN air electrode operated at 3% humidified H₂O at 600 °C.

Supplementary Figure S17 Detailed SEM images of BCFN air electrode after operated under 3% humidified air for 10 hours at (a) room temperature, (b) 400 °C, (c) 450 °C, (d) 500 °C, (e) 550 °C and (f) 600 °C, respectively. (g) Typical SEM images of BCFN air electrode after operated with 3% H₂O humidified air at 600 °C for 10 hours and followed operated with dry air at 600 °C for 10 hours. Temperature dependent exsolved nanoparticles population of BCFN air electrode operated at 3% H₂O humidified air for 10 hours. (h) populations of the exsolved nanoparticles with diameter of 50-100 nm (blue) and over 100 nm (purple) when BCFN electrode was operated at 3% humidified H₂O for 10 hours at 400-600 °C.

In order to investigate the specific temperature that nanoparticles start to exsolve from the skeleton, we operated the BCFN air electrode at different temperatures (400-600 °C) with 3% humidified air for 10 hours at OCV condition (Figure S17). It is revealed that BCFN approximately started to exsolve at 400 °C. Similarly, we roughly estimated the population of the exsolved nanoparticles (NP) and found that the NP population (over 100nm) increased significantly as temperature increased.

Supplementary Figure S18. Typical SEM images of BCFN air electrode after operated at OCV condition with wet (3% H₂O) hydrogen as fuel and dry air as oxidant (a) & (d), a current density of -0.5 A cm⁻² with wet (3% H₂O) hydrogen as fuel and dry air as oxidant (b) & (e), and a current density of +0.5 A cm⁻² with wet (3% H₂O) hydrogen as fuel and dry air as oxidant (c) & (f) at 600 °C for 10 hours; Typical SEM images of BCFN air electrode after operated at OCV condition with wet (3% H₂O) hydrogen as fuel and wet (3% H₂O) air as oxidant (g) & (j), a current density of -0.5 A cm⁻² with wet (3% H₂O) hydrogen as fuel and wet (3% H₂O) air as oxidant (h) & (k) and a current density of +0.5 A cm⁻² with wet (3% H₂O) hydrogen as fuel and wet (3% H₂O) air as oxidant (i) & (l) at 600 °C for 10 hours.

To evaluate if the voltage/current or the chemical potential of steam is the main reason for the exsolution, we treated the air electrode under the following conditions. First, we operated cells with BCFN air electrode at OCV condition when the air electrode is exposed to dry air at 600 °C for 10 hours at current densities of 0, -0.5 and +0.5 A cm⁻². It is shown that very few nanoparticles are exsolved (Figure S18a-f). In contrast,

when the air electrode is exposed to wet air (3% H₂O) at current densities of 0, -0.5, and +0.5 Acm⁻², a great number of nanoparticles were observed on the skeleton (Figure S18 g-1), indicating that steam can lead to the exsolution. Therefore, it is reasonable to conclude that the current flowing may not be the factor required to induce the exsolution. The steam is most likely the main reason for the exsolution.

Reviewer #3:

Recently, reversible protonic ceramic electrochemical cells (PCECs) have emerged as a new and efficient energy storage and conversion device. This work focused on addressing the tough challenges of the air electrode: the poor performance and insufficient stability of the air electrode. Overall, the experiments have been designed and performed nicely. The excellent cell performance (peak power density in fuel cell mode, current density at an electrolysis voltage) and demonstration of stability for over 300 hours warrant publication. The manuscript is also well organized. I believe that the results reported in this manuscript are of big interest to the broad readership of Nat Commun. The manuscript can be recommended for publication in Nat Commun after some minor revisions. The detailed comments/suggestions are shown below:

Response to general comments: We very much appreciate the positive comments from Reviewer #3.

Comment #1 of Reviewer 3

The mass transfer is very important, especially when the cell is in the electrolysis mode. What is the pore structure of the cells fabricated for this study? The authors need to provide such information, at least the SEM image with more details of the fuel electrode and the air electrode.

Response to C1: We very much appreciate the valuable comments from Reviewer #3.

More details about the air electrode, electrolyte, anode functional layer and anode support layer can be found in **Figure R10**. The fuel electrode of a full cell often consists of a functional layer (~25-30 μm) and a supporting layer (~600 μm) (**Figure R10**). The functional layer (with less and finer pores) is expected to provide more triple phase boundaries and larger surface area for electrochemical reactions, while the supporting layer (with more and larger pores) is expected to provide facile paths for mass transport. We have added the related pore information of electrodes to the revised manuscript. The information of the air electrode can be found in the inset of Figure 3b.

Figure R10. Typical SEM image of a single cell with distinct four layers.

Revision: We have added the related information to the revised manuscript.

As shown, a multi-layered structure, including a porous Ni-BZCYYb anode supporting layer (ASL, ~600 μm), a porous but fine Ni-BZCYYb anode functional layer (AFL, ~25 μm), a dense BZCYYb electrolyte layer (~10 μm), and a porous BCFN layer (~15 μm), were adhered well with no cracks or delamination

(Supplementary Fig. S4). The functional layer has finer pores and a larger surface area (due mainly to the reduction of NiO), providing more triple-phase boundaries for electrochemical reactions. Meanwhile, the supporting layer has larger pores and continuous channels (due mainly to the removal of pore-former), providing facile paths for mass transport.

Supplementary Figure S4. Typical SEM image of a single cell with distinct four layers.

Comment #2 of Reviewer 3

Insufficient information about the cell testing, such as the active surface area of the cells, flow rates of gases, is provided. That information is very important when evaluating the cell Faradaic efficiency. In addition, more details about the FE measurement should be added since a small cell area would lead to large uncertainty.

Response to C2: For most of the tests, button cells with an effective area of 0.28 cm^2 were used. For the Faradic efficiency test, 1-inch cells (shown in **Figure R14**) with an effective area of 2 cm^2 were used to increase the accuracy of the measurement. For the fuel cell test, 20 mL min^{-1} humidified H_2 (3% H_2O) was supplied to the fuel electrode as the fuel and air in the air electrode as the oxidant. For the electrolysis test, the fuel electrode was fed with 20 mL min^{-1} humidified H_2 (3% H_2O) and the air electrode was exposed to 200 mL min^{-1} humidified (3% H_2O) air. Faradic efficiencies

(η_F) were measured based on the ratio of the experimental and theoretical hydrogen generation amounts at fixed current densities. 40 mL min⁻¹ 20% H₂-80% N₂ was fed to the fuel electrode and 60 mL min⁻¹ humidified air was fed to the air electrode. Gas chromatography was used to monitor the hydrogen concentration in the fuel electrode, which was used to calculate the amount of actual hydrogen generated. The steam concentration was controlled by a humidification system (Fuel Cell Technologies, Inc.).

Figure 11 Photograph of 1-inch cells for Faradaic efficiency measurement

Revision: We have added those details in the experimental section in the revised manuscript.

“For most of the tests, button cells with an effective area of 0.28 cm² were used. For the Faradic efficiency test, 1-inch cells with an effective area of 2 cm² were used to increase the accuracy of the measurement. For the fuel cell test, the fuel electrode was exposed to 20 mL min⁻¹ humidified H₂ (3% H₂O) and the air electrode was exposed to ambient air. For the electrolysis test, the fuel electrode was fed with 20 mL min⁻¹ humidified H₂ (3% H₂O) and the air electrode was exposed to 200 mL min⁻¹ humidified (3% H₂O) air. The steam concentration was controlled by a humidification system (Fuel Cell Technologies, Inc.). Faradic efficiencies (η_F) were measured based on the ratio of the experimental and theoretical hydrogen generation amounts at fixed current densities. 40 mL min⁻¹ 20% H₂-80% N₂ was fed to the fuel electrode and 60 mL min⁻¹ humidified air was fed to the air electrode. Gas chromatography was used to

monitor the hydrogen concentration in the fuel electrode, which was used to calculate the amount of actual hydrogen generated.”

Comment #3 of Reviewer 3

The most interesting point of this work is the surface re-structuring of the air electrode. However, such information is not provided in detail. What temperature did those nanoparticles start to exsolve? Is the exsolution process reversible?

Reply to C3: We thank the reviewer for this insightful comment.

In order to address the concerns, we have performed additional experiments to clarify the surface re-structuring of the air electrode, including treating the BCFN samples at different treating temperatures, and different treating times at 600°C. We also evaluate if the voltage/current or the chemical potential of steam is the main reason for the exsolution.

First, we operated the BCFN air electrode at 600 °C in the air with 3% H₂O for different durations (5-40 hours) at OCV conditions to disclose the relationship between the amount of exsolved nanoparticles and operating time (**Figure R10**). As shown, a considerable number of nanoparticles (with a population of ~20 μm⁻²) have been exsolved after 5 hours of treatment. The precipitated nanoparticles significantly increase as a function of treating time (in the following 35h). However, after 10h, the NP population tends to be stable.

Figure R10. Detailed SEM images of BCFN air electrode after operated under 3% humidified air at 600 °C for 5 hours (a) & (d), 10 hours (b) & (e), 20 hours (c) & (f), 30 hours (g) & (i), 40 hours (h) & (j), respectively. (k) Time dependence of exsolved nanoparticles populations of BCFN air electrode operated at 3% humidified H₂O at 600 °C.

Figure R11 Detailed SEM images of BCFN air electrode after operated under 3% humidified air for 10 hours at (a) room temperature, (b) 400 °C, (c) 450 °C, (d) 500 °C, (e) 550 °C and (f) 600 °C, respectively. (g) Typical SEM images of BCFN air electrode after operated with 3% H₂O humidified air at 600 °C for 10 hours and followed operated with dry air at 600 °C for 10 hours. Temperature dependent exsolved nanoparticles population of BCFN air electrode operated at 3% H₂O humidified air for 10 hours. (h) populations of the exsolved nanoparticles with diameter of 50-100 nm (blue) and over 100 nm (purple) when BCFN electrode was operated at 3% humidified H₂O for 10 hours at 400-600 °C.

In order to investigate the specific temperature that nanoparticles start to exsolve from the skeleton, we operated the BCFN air electrode at different temperatures (400-600 °C) with 3% humidified air for 10 hours at OCV condition (**Figure R11**). It is revealed that BCFN approximately started to exsolve at 400 °C. Similarly, we roughly estimated the population of the exsolved nanoparticles (NP) and found that the NP population (over 100nm) increased significantly as temperature increased.

To identify whether the Nb-deficient BCFN nanoparticle exsolution process is reversible, we operated a single cell with BCFN air electrode in wet air with 3% H₂O wet air for 10 hours at

600°C to induce the nanoparticles. The flow rate of air was kept at 20 mL min⁻¹ during the testing. The SEM images of the BCFN surface are shown in **Figure R11f**. As expected, a significant number of nanoparticles was observed. We operated another cell with a BCFN air electrode when exposed to wet air for 10h, and then to dry air for another 10 hours at 600 °C. The flow rate of air was kept at 20 mL min⁻¹ during the testing. Shown in **Figure R11g** is the SEM image of the BCFN air electrode after the above treatment. It is demonstrated that the morphology and numbers of exsolved nanoparticles on the surface are similar to those observed in **Figure R11f**, indicating that the NPs induced by steam cannot return to the bulk. The exsolution process is more likely irreversible.

Figure R12. Typical SEM images of BCFN air electrode after operated at OCV condition with wet (3% H₂O) hydrogen as fuel and dry air as oxidant (a) & (d), a current density of -0.5 A cm⁻² with wet (3% H₂O) hydrogen as fuel and dry air as oxidant (b) & (e), and a current density of +0.5 A cm⁻² with wet (3% H₂O) hydrogen as fuel and dry air as oxidant (c) & (f) at 600 °C for 10 hours; Typical SEM images of BCFN air electrode after operated at OCV condition with wet (3% H₂O) hydrogen as fuel

and wet (3% H₂O) air as oxidant **(g)** & **(j)**, a current density of -0.5 A cm⁻² with wet (3% H₂O) hydrogen as fuel and wet (3% H₂O) air as oxidant **(h)** & **(k)** and a current density of +0.5 A cm⁻² with wet (3% H₂O) hydrogen as fuel and wet (3% H₂O) air as oxidant **(i)** & **(l)** at 600 °C for 10 hours.

To evaluate if the voltage/current or the chemical potential of steam is the main reason for the exsolution, we treated the air electrode under the following conditions. First, we operated cells with BCFN air electrode at OCV condition when the air electrode is exposed to dry air at 600 °C for 10 hours at current densities of 0, -0.5 and +0.5 Acm⁻². It is shown that very few nanoparticles are exsolved (**Figure R12a-f**). In contrast, when the air electrode is exposed to wet air (3% H₂O) at current densities of 0, -0.5, and +0.5 Acm⁻², a great number of nanoparticles were observed on the skeleton (Figure R12 g-l), indicating that steam can lead to the exsolution. Therefore, it is reasonable to conclude that the current flowing may not be the factor required to induce the exsolution. The steam is most likely the main reason for the exsolution.

Revision: We have added the above discussion in the revised Supporting Information.

First, we operated the BCFN air electrode at 600 °C in the air with 3% H₂O for different durations (5-40 hours) at OCV conditions to disclose the relationship between the amount of exsolved nanoparticles and operating time (**Figure S16**). As shown, a considerable number of nanoparticles (with a population of ~20 μm⁻²) have been exsolved after 5 hours of treatment. The precipitated nanoparticles significantly increase as a function of treating time (in the following 35h). However, after 10h, the NP population tends to be stable.

Supplementary Figure S16. Detailed SEM images of BCFN air electrode after operated under 3% humidified air at 600 °C for 5 hours (a) & (d), 10 hours (b) & (e), 20 hours (c) & (f), 30 hours (g) & (i), 40 hours (h) & (j), respectively. (k) Time dependence of exsolved nanoparticles populations of BCFN air electrode operated at 3% humidified H₂O at 600 °C.

Supplementary Figure S17 Detailed SEM images of BCFN air electrode after operated under 3% humidified air for 10 hours at (a) room temperature, (b) 400 °C, (c) 450 °C, (d) 500 °C, (e) 550 °C and (f) 600 °C, respectively. (g) Typical SEM images of BCFN air electrode after operated with 3% H₂O humidified air at 600 °C for 10 hours and followed operated with dry air at 600 °C for 10 hours. Temperature dependent exsolved nanoparticles population of BCFN air electrode operated at 3% H₂O humidified air for 10 hours. (h) populations of the exsolved nanoparticles with diameter of 50-100 nm (blue) and over 100 nm (purple) when BCFN electrode was operated at 3% humidified H₂O for 10 hours at 400-600 °C.

In order to investigate the specific temperature that nanoparticles start to exsolve from the skeleton, we operated the BCFN air electrode at different temperatures (400-600 °C) with 3% humidified air for 10 hours at OCV condition (Figure S17). It is revealed that BCFN approximately started to exsolve at 400 °C. Similarly, we roughly estimated the population of the exsolved nanoparticles (NP) and found that the NP population (over 100nm) increased significantly as temperature increased.

Supplementary Figure S18. Typical SEM images of BCFN air electrode after operated at OCV condition with wet (3% H₂O) hydrogen as fuel and dry air as oxidant (a) & (d), a current density of -0.5 A cm⁻² with wet (3% H₂O) hydrogen as fuel and dry air as oxidant (b) & (e), and a current density of +0.5 A cm⁻² with wet (3% H₂O) hydrogen as fuel and dry air as oxidant (c) & (f) at 600 °C for 10 hours; Typical SEM images of BCFN air electrode after operated at OCV condition with wet (3% H₂O) hydrogen as fuel and wet (3% H₂O) air as oxidant (g) & (j), a current density of -0.5 A cm⁻² with wet (3% H₂O) hydrogen as fuel and wet (3% H₂O) air as oxidant (h) & (k) and a current density of +0.5 A cm⁻² with wet (3% H₂O) hydrogen as fuel and wet (3% H₂O) air as oxidant (i) & (l) at 600 °C for 10 hours.

To evaluate if the voltage/current or the chemical potential of steam is the main reason for the exsolution, we treated the air electrode under the following conditions. First, we operated cells with BCFN air electrode at OCV condition when the air electrode is exposed to dry air at 600 °C for 10 hours at current densities of 0, -0.5 and +0.5 Acm⁻².

It is shown that very few nanoparticles are exsolved (**Figure S18a-f**). In contrast, when the air electrode is exposed to wet air (3% H₂O) at current densities of 0, -0.5, and +0.5 Acm⁻², a great number of nanoparticles were observed on the skeleton (**Figure S18 g-l**), indicating that steam can lead to the exsolution. Therefore, it is reasonable to conclude that the current flowing may not be the factor required to induce the exsolution. The steam is most likely the main reason for the exsolution.

REVIEWERS' COMMENTS

Reviewer #1 (Remarks to the Author):

While LEIS would have been a really good evidence, most of the comments are addressed satisfactorily. Manuscript may be considered for publication in the present form.

Reviewer #2 (Remarks to the Author):

The authors have satisfactorily addressed the reviewer's comments. The revised manuscript is acceptable for publication in this journal.

Point-to-point Responses to Reviewers

First of all, we very much appreciate the reviewers' time and efforts spent on reviewing our manuscript in the first round. The Reviewers' thoughtful comments and suggestions have significantly improved the quality and clarity of our manuscript. We very much appreciate the reviewers' comments on our manuscript. Our point-to-point responses in the second-round review are presented below.

Reviewer(s)' Comments to Author:

Reviewer #1: (Remarks to the Author):

While LEIS would have been a really good evidence, most of the comments are addressed satisfactorily. Manuscript may be considered for publication in the present form.

Reply to the comments: We very much appreciate the suggestion from Reviewer #1.

Reviewer #2 (Remarks to the Author): The authors have satisfactorily addressed the reviewer's comments. The revised manuscript is acceptable for publication in this journal.

Reply to the comments: We very much appreciate the suggestion from Reviewer #2.